# ORCHESTRATING PRE-TRAINED AGENTS FOR MULTI-OBJECTIVE DECISION MAKING

## ABSTRACT

Multi-objective sequential decision making (MO-SDM) is central to many real-world tasks, where an agent must make a sequence of decisions that balance multiple, often conflicting objectives. Multi-objective reinforcement learning (MORL) is a common approach to solving MO-SDM problems, but typically requires training from scratch under known objective configurations. In this paper, we propose a zero-shot paradigm: reusing a set of existing or pre-trained single-objective (SO) policies through large language model (LLM)-driven orchestration. We formalize this setting using context engineering and develop three types of orchestrators that vary in the context components they observe, such as *knowledge*, *tools*, and *reflection*, and in their ability to reason over policy behavior. Experiments on two domains (education and control) demonstrate that our method achieves competitive Pareto quality and per-objective performance while reducing computational cost by over $3\times$ compared to MORL methods. An ablation study further reveals how context richness and reflective foresight influence zero-shot decision quality.

## 1 INTRODUCTION

Sequential decision making is central to many real-world applications. For example, an educational platform recommends a sequence of tests to improve a student's long-term learning outcomes (Szpektor et al., 2013), and a mobile robot plans trajectories that simultaneously minimize energy consumptions and maximize safety in a dynamic environment (Chevalier-Boisvert et al., 2023). These scenarios often involve *multiple, potentially conflicting objectives*, such as balancing performance and aptitude in Education, or trading off efficiency and robustness in control, making the problem of *multi-objective sequential decision making* (MO-SDM) both ubiquitous and challenging (Deb et al., 2016; Gunantara, 2018).

Reinforcement learning (RL) has achieved tremendous success in sequential decision-making tasks. However, most RL algorithms assume a *single fixed objective* (Sutton et al., 1998; Roijers et al., 2013), as optimizing a scalar reward simplifies theoretical analysis and algorithmic design (Mnih et al., 2015; Schulman et al., 2017). To address multi-objective scenarios, prior work in multi-objective RL (MORL) typically trains a family of policies across objective preferences, either via scalarization (Roijers et al., 2013; Icarte et al., 2018), preference-conditioned networks (Abels et al., 2019), Pareto-front approximators (Van Moffaert & Nowé, 2014; Reymond & Nowé, 2019), or gradient-based methods (Désidéri, 2009; Yu et al., 2020). More related work is discussed in Appendix A.1.

These methods, however, assume that the full set of objectives is known *a priori* during training, which is rarely the case in practice (Hayes et al., 2021). For instance, a household robot may encounter emerging goals as task complexity increases, such as avoiding hazards like spilled liquids or retrieving specific items. Scalarized or fixed-policy methods must then be redesigned and retrained (Yu et al., 2020; Van Moffaert & Nowé, 2014; Roijers et al., 2013), which is prohibitively expensive given the unknown and unbounded nature of the objective space. Moreover, objectives often vary across contexts. In education applications, high-performing students may benefit from questions that maximize learning potential, while others may require content that closes knowledge gaps and maintains engagement (Vygotsky, 1987; Davis, 1977; Wang et al., 2023; Li et al., 2021; Abdelrahman et al., 2023). These limitations make traditional MORL methods ill-suited for deployment scenarios where objective configurations are uncertain or may shift at inference time.

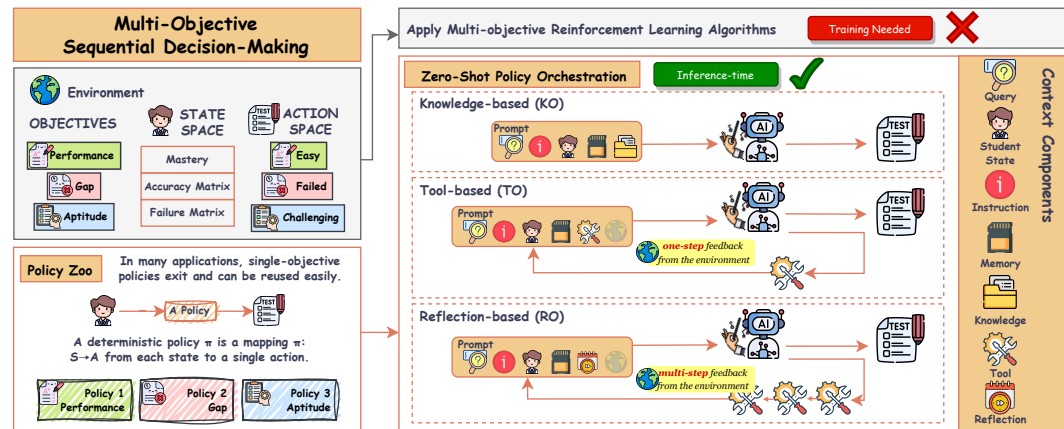

Figure 1: **Overview of the zero-shot policy orchestration framework for MO-SDM.** Instead of training a new MORL policy given objectives, we reuse existing SO policies in the policy zoo via LLM-based policy orchestration. Three orchestrators (KO, TO, RO) differ in how they incorporate structured context and environment feedback.

In many real-world deployments, especially in safety-critical or user-facing applications, it is common to maintain a collection of single-objective (SO) policies, each optimized for a specific goal. These policies form a readily available *policy zoo*. Rather than discarding them, it is desirable to *reuse and orchestrate* them to meet new, multi-objective task requirements without retraining a new MORL policy (Nikookar et al., 2025). The key challenge is: *how can we reuse SO policies to satisfy novel objective trade-offs at inference time?*

Recent advances in large language models (LLMs) have shown that they can serve as powerful decision-making agents capable of reasoning, reflection, and tool use (Shinn et al., 2023; Zhou et al., 2023; Yao et al., 2022; Wei et al., 2022). Context engineering (Mei et al., 2025) further formalizes how structured prompts, comprising knowledge, memory, feedback, and more components, shape LLM reasoning. We hypothesize that, using *context engineering* (Mei et al., 2025), i.e., encoding goals, preferences, or user traits into structured prompts or memory contexts, LLMs can be used to orchestrate SO policies in a zero-shot fashion to solve MO-SDM problems at inference time.

Within this problem space, we focus on two research questions. **RQ1:** Do zero-short policy orchestration approaches, powered by LLMs and context engineering, compete with training-based MORL approaches in MO-SDM tasks? **RQ2:** which context components most strongly contribute to the orchestration capabilities? Our contributions are fivefold:

1. We formalize MO-SDM as a context-driven inference-time problem based on context engineering, enabling zero-shot policy orchestration over a pre-trained SO policy zoo without retraining.

2. We develop three orchestration variants that systematically vary in context richness, black-box policy invocation, and reflective foresight: **Knowledge-based Orchestrator (KO)** uses static policy descriptions to reason over goals; **Tool-based Orchestrator (TO)** interacts with policies as black-box tools via one-step rollouts; and **Reflection-based Orchestrator (RO)** performs multi-step lookahead to anticipate long-term effects.

3. We conduct the first methodological comparison between LLM-based zero-shot policy orchestration and RL-based MO training pipelines on two domains (education and control), revealing trade-offs in decision quality and cost-efficiency.

4. We perform a fine-grained ablation study to identify which context components most influence the orchestration performance.

5. We provide empirical evidence that our orchestration methods match or outperform MORL methods while reducing computational cost by over $3\times$.

## 2 BACKGROUND AND PROBLEM STATEMENT

We first review the formulation of MO-SDM under a deterministic Markov Decision Process (MDP) and then formally define the problem setting addressed in this work: zero-shot policy orchestration.

## 2.1 Multi-Objective Markov Decision Processes (MO-MDPs)

**Optimization Objectives.** We consider a set of $m$ objectives $\mathcal{D} = \{d_1, d_2, \ldots, d_m\}$, where each dimension $d_i$ captures a distinct optimization goal. In Education, objectives could be maximizing performance, minimizing failures (aka, gap reduction), or maximizing learning potential (aka, aptitude).

**MO-MDP.** We model the environment as an *MO-MDP*, defined as a tuple $\mathcal{M} = \langle \mathcal{S}, \mathcal{A}, \{\mathcal{R}_i\}_{i=1}^m, P, H \rangle$, where $\mathcal{S}$ is the state space, $\mathcal{A}$ the action space, $P(s'|s, a)$ is the transition function and $H \in \mathbb{N}$ is the finite horizon. Each objective $d_i$ is associated with a reward function $\mathcal{R}_i : \mathcal{S} \times \mathcal{A} \times \mathcal{S} \rightarrow \mathbb{R}$. At every step, the agent receives a vector-valued reward $\mathbf{r}(s, a, s') = \big(\mathcal{R}_1(s, a, s'), \ldots, \mathcal{R}_m(s, a, s')\big) \in \mathbb{R}^m$..

In Education, $\mathcal{S}$ may represent the student's skill mastery and question history. $\mathcal{A}$ corresponds to question selection strategies, either from easy ones, failed ones, or high-aptitude ones. $\mathcal{R}$ captures diverse reward signals, such as correctness (performance), targeting known skill gaps (gap reduction), and attempting questions above current mastery level (aptitude).

**Policy and Return.** A policy $\pi : \mathcal{S} \times \mathcal{A} \rightarrow [0, 1]$ defines the probability of taking action $a \in \mathcal{A}$ in state $s \in \mathcal{S}$, $\pi(s, a) = \Pr(a_t = a | s_t = s)$. The *cumulative return* under $\pi$ for objective $d_i$ over horizon $H$ is $G^{d_i}(\pi) = \sum_{t=0}^{H-1} \mathcal{R}_i(s_t, a_t, s_{t+1})$, where the trajectory $(s_0, a_0, \ldots, s_H)$ is generated by following $\pi$. We denote the full reward vector as $\mathbf{G}(\pi) = \big(G^{d_1}(\pi), \ldots, G^{d_m}(\pi)\big)$.

**Pareto Optimality.** A policy $\pi$ is *Pareto-dominated* if there exists a policy $\pi'$ such that $G^{d_i}(\pi') \geq G^{d_i}(\pi)$ for all $i$, and $G^{d_j}(\pi') > G^{d_j}(\pi)$ for some $j$. The set of non-dominated policies forms the *Pareto frontier*, which offers a formal ideal of optimal trade-offs among conflicting objectives. This refines our key challenge stated in the introduction: *how can we reuse SO policies to find solutions that are close to the Pareto frontier at inference time, without retraining?*

## 2.2 Problem Statement: Policy Orchestration

We assume access to a set of $m$ pre-trained SO policies $\Pi = \{\pi_1, \pi_2, \ldots, \pi_m\}$, all defined over the same deterministic MDP. Each policy $\pi_k$ is trained to optimize a single objective $d_i \in \mathcal{D}$, using only the corresponding reward component $\mathcal{R}_i$.

All policies in $\Pi$ expose a unified inference interface: given a state $s_t$, they return an action $a_t \sim \pi_k(\cdot \mid s_t)$. This common interface makes policies modular and interchangeable, enabling a higher-level *orchestrator* to query, compare, or compose them at inference time.

Our zero-shot policy orchestration problem seeks a policy $\pi_{\mathcal{O}}$, induced by an *orchestrator* $\mathcal{O}$, that achieves favorable MO trade-offs, i.e., close to the Pareto frontier, across $\mathcal{D}$ without retraining.

## 3 Proposed Solution

We propose a zero-shot policy orchestration framework for MO-SDM tasks as shown in Fig. 1. In question recommendation, we seek to enable student upskilling by recommending $k$ sets of questions that improve performance, reduce gaps, and increase aptitude. Given three SO policies, an MO-SDM problem over all objectives would be solved by orchestrating among them at inference time, thereby avoiding training an MO policy.

### 3.1 Context-Driven Formalization of Policy Orchestration

We now formalize the zero-shot policy orchestration problem stated in Sec. 2.2 under the framework of LLMs. Specifically, we cast the decision of selecting the optimal policy $\pi^*$ as an inference-time prediction by an autoregressive LLM, parameterized by $\theta$. Given a structured context $C$, the LLM outputs a policy index from the candidate set $\Pi$. Our objective is to construct a context $C$ that maximizes the probability of selecting the optimal policy:

$$\max_C P_\theta(\pi^* \mid C).$$

Following the principles of context engineering (Mei et al., 2025), we define the context $C$ as an assembly of components: $C = \mathcal{A}(c_1, c_2, \ldots, c_n)$, where each $c_i$ is a context component (e.g., a system instruction, environment description, task metadata, or tools), and $\mathcal{A}$ is an assembly function that structures these components into a prompt.

We describe the components $c_i$ that we use. Figures 2 and 3 show their prompt templates and the detailed prompts are in Appendix A.2.

- $c_{\text{query}}$: The immediate MO-SDM task request, i.e., "recommend $k$ questions to the student...".
- $c_{\text{state}}$: The dynamic state of the system. This captures current mastery, test successes, and failures in question recommendation.
- $c_{\text{instr}}$: System instructions and rules describing the orchestrator's role and environment (Figure 2 left), i.e., "You are a MO policy orchestrator...".
- $c_{\text{mem}}$: Persistent information from prior interactions (Figure 2 right), i.e., the student's historical performance and mastery trajectories. We distinguish *short-term memory* $c_{\text{mem-s}}$, which records only the most $l$ recent interactions (where $l \in \mathbb{Z}$, $l \geq 1$), from *long-term memory* $c_{\text{mem-l}}$, which summarizes the entire learning trajectory.

```
- Role: you are a MO policy
↪   orchestrator...
- Task: select a policy for the next
↪   action
- Guardrails
- Instructions
```

```
=== SHORT-TERM MEMORY ===
- Current state
- Last selected policy and action
- Immediate outcome (reward, feedback)
- Per-objective performance signals
=== LONG-TERM MEMORY ===
- Trends in environment or user behavior
- Policy usage patterns
- Strategies linked to successful learning
```

Figure 2: Context components $c_{\text{instr}}$ (left) and $c_{\text{mem}}$ (right).

- $c_{\text{know}}$: External knowledge typically retrieved via RAG [1] or knowledge graphs (Figure 3 left). In our problem, this corresponds to natural-language descriptions of policies in $\Pi$.
- $c_{\text{tools}}$: Definitions and signatures of available external tools (Figure 3 right), i.e., the executable policies in $\Pi$.

```
Each policy is annotated with:
- its optimized objective
- high-level behavior profile
- typical use cases and failure modes
- example trajectories or outputs
```

```
Each policy is exposed as a tool with:
- its optimized objective
- input signature (state features)
- output signature (action or
↪   recommendation)
```

Figure 3: Context components $c_{\text{know}}$ (left) and $c_{\text{tools}}$ (right).

## 3.2 PROPOSED ORCHESTRATION SOLUTIONS

Our solution addresses the MO-SDM problem by making a sequence of policy selection decisions. At each time step $t$, it receives a structured context $C_t$ and selects the most appropriate SO policy $\pi_k \in \Pi$ for the current decision. We design three orchestrator variants that differ in the context components they observe, corresponding to increasing levels of decision-time information and reasoning capability, as summarized in Table 1.

**Knowledge-based Orchestrator (KO).** KO receives a context $C_{\textbf{KO}}$ containing including policy descriptions $c_{\text{know}}$, the current system state $c_{\text{state}}$, and memory $c_{\text{mem}}$ (short- and long-term). Each policy in $\Pi$ is described via its training objective, high-level behavior, and representative outputs. This setup enables zero-shot generalization to new states by leveraging static knowledge. Formally, $C_{\text{KO}} = \{c_{\text{query}}, c_{\text{state}}, c_{\text{instr}}, c_{\text{mem}}, c_{\text{know}}\}$.

**Tool-based Orchestrator (TO).** Unlike **KO**, which relies on textual policy descriptions, **TO** treats each $\pi_k \in \Pi$ as a black-box tool callable via its function signature in $c_{\text{tools}}$. At time $t$, TO performs a *one-step look-ahead*, invoking up to $k_{\text{TO}}$ candidate policies to simulate next-step outcomes given

---

[1]RAG: Retrieval Augmented Generation

Table 1: Context components used by different orchestrators.

| Orchestrator | $c_{query}$ | $c_{state}$ | $c_{instr}$ | $c_{mem}$ | $c_{know}$ | $c_{tools}$ | $c_{reflct}$ |
|---|---|---|---|---|---|---|---|
| Knowledge-based (KO) | ✓ | ✓ | ✓ | ✓ | ✓ | ✗ | ✗ |
| Tool-based (TO) | ✓ | ✓ | ✓ | ✓ | ✗ | ✓ | ✗ |
| Reflection-based (RO) | ✓ | ✓ | ✓ | ✓ | ✗ | ✓ | ✓ |

the current state. This allows TO to reason over policy behaviors based on environment feedback rather than static metadata, making it suitable for large or opaque policy sets. Formally, $C_{\mathbf{TO}} = C_{\mathbf{KO}} - c_{know} + c_{tools}$.

This design follows the reasoning–feedback–refinement paradigm commonly seen in decision-time adaptation frameworks such as *Reflexion* (Shinn et al., 2023). Specifically, TO performs a loop over three roles: (1) acting by selecting a policy to probe; (2) evaluating by executing the policy for one step and receiving environment feedback; (3) refining by integrating the feedback into $c_{mem}$ to inform subsequent decisions. This structure enables lightweight self-improvement without additional training, and is particularly effective when policy internals are inaccessible.

**Reflection-based Orchestrator (RO). RO** extends **TO** by incorporating deeper foresight. Rather than probing policies one step ahead, RO performs $\ell$-*step look-ahead rollouts* for $k_{RO}$ candidate policies in parallel. This simulates how the environment may evolve under such policy candidate and allow RO to anticipate long-term effects of such choice. We introduce $c_{reflct}$ as a new context component. It records the simulated rollout's future state, actions, and per-objective rewards. Inspired by the *Language Agent Tree Search (LATS)* framework (Zhou et al., 2023), RO selects a policy that balances short-term gains with long-term trade-offs. Formally, $C_{\mathbf{RO}} = C_{\mathbf{TO}} + c_{reflct}$.

This design comes with a clear *trade-off*: while **RO** may discover better or more robust policies by exploring multiple paths, it incurs higher inference cost due to multi-step and multi-branch simulation. We explicitly evaluate it in Sec. 4.2.1.

All orchestrators use Chain-of-Thought (CoT) prompting by default: the LLM is instructed to produce explicit intermediate reasoning steps before selecting a policy. In **TO**, CoT guides reasoning over tool outputs and recent history; in **RO**, it unfolds parallel reflective paths.

Overall, KO, TO, and RO span a design spectrum: (1) **KO** uses static policy descriptions to reason over goals; (2) **TO** interacts with policies as black-box tools via one-step rollouts; (3) **RO** performs multi-step lookahead to anticipate long-term effects. This spectrum allows us to study how context richness, black-box invocation, and reflective rollouts influence MO decision quality.

## 4 EXPERIMENTS

### 4.1 EXPERIMENTAL SETUP

**Education.** We consider a *question recommendation* environment in the education domain, which recommends a sequence of test questions to improve student learning outcomes. This is an MO-SDM setting optimizing aptitude ($d_1$), gap reduction ($d_2$), and performance ($d_3$). We use real-world question sets from MathE[2], an e-learning platform for enhancing mathematical skills in higher education. MathE contains 1,913 questions across 15 topics, each with a discrete difficulty scale ranging from 1 to 6. The environment simulates student responses using the Item Response Theory (IRT) (Embretson & Reise, 2013). The MDP formalization is provided in Appendix A.4.

**MiniGrid.** MiniGrid (Chevalier-Boisvert et al., 2023) is a practical multi-task control challenge with conflicting implicit goals (e.g., reaching the goal ($d_1$), avoiding lava ($d_2$), collecting gold ($d_3$)). We define multi-objective environments from MO-Gymnasium with a 6×6 grid size. We give a detailed MDP formalization in Appendix A.5.

#### 4.1.1 EVALUATION.

We evaluate both quantitative and qualitative aspects of MO decision quality.

---

[2]MathE: https://mathe.ipb.pt/

- *Episode efficiency.* We measure the average episode length.
- *Pareto front quality.* We use **hypervolume** (HV) to measure the trade-off coverage and **sparsity** to measure the diversity of solutions across objective preferences.
- *Objective-specific metrics.* In Education, we report the average normalized cumulated reward for each objective (aptitude, gap, performance), denoted as $R_{d_1}$, $R_{d_2}$, and $R_{d_3}$. For MiniGrid, we report *Success Rate (%)*, *Gold Collection Rate (%)*, and *Lava Collision Rate (%)*.
- *Computational Cost.* We report both training and inference monetary costs and carbon footprints to compare the efficiency of retraining-based (MORL) and zero-shot (orchestration) methods. Full formulas and results are provided in the Appendix A.6.
- *Mastery progression (Education).* We track students' mastery levels over steps, which captures how different policies accelerate or delay learning progress.
- *Flow alignment (Education).* Based on the Flow Theory and the Zone of Proximal Development (ZPD) (Nakamura & Csikszentmihalyi, 2014; Davis, 1977; Vygotsky, 1987; Bouarour et al., 2024), we visualize the distribution of recommended questions in the mastery–difficulty space to assess if questions remain in the flow zone (neither under- nor over-challenging the student).
- *Qualitative measures (Education).* We further compare orchestrators' trajectories and assess subjective dimensions such as mental demand and frustration using an LLM-based judge.

### 4.1.2 POLICY ZOO.

For each application, we construct a set of SO policies using PPO from Stable-Baselines3 (Schulman et al., 2017; Raffin et al., 2021), each optimized toward a distinct objective (**SO-**$d_i$). In the education domain, SO policies are trained in the same environment with objective-specific rewards. In contrast, MiniGrid policies are trained in separate sub-environments, each containing only the elements relevant to a specific objective (e.g., gold, lava, or goal). This avoids interference between objectives during training and ensures each policy exhibits pure SO behavior.

### 4.1.3 REFERENCE MO POLICIES.

To provide upper bounds on achievable MO performance, we train dedicated MO policies as reference baselines. In the education domain, we implement three variants: (1) **MO-S**, standard PPO with fixed linear scalarization of objectives (Schulman et al., 2017), (2) **MO-RM**, standard PPO with reward machine (Icarte et al., 2018), and (3) **MO-MGDA**, integrating Multiple Gradient Descent Algorithm (MGDA) into PPO updates (Désidéri, 2009). In contrast, MiniGrid does not expose reward vectors during training. Following standard practice (Chevalier-Boisvert et al., 2023), we train a PPO policy directly on the multi-objective environment using a shaped scalar reward that reflects multiple objectives (reaching the goal (+1), collecting gold (+1 each), and implicitly avoiding lava (no reward)). We name it **MO-pseudoS**, a *pseudo-scalarized* MO policy. More explanations about the RL training are in Appendix A.3

### 4.1.4 ORCHESTRATORS.

We evaluate our orchestrators defined in Sec. 3: **KO**, **TO**, **RO**. We instantiate them with three LLMs: LLaMA-3-8B[3], Mistral-24B[4], and Claude-3.7-Sonnet[5] (reasoning-enabled).

## 4.2 RESULTS

### 4.2.1 **RQ1:** HOW DO ZERO-SHOT POLICY ORCHESTRATION APPROACHES PERFORM COMPARED TO TRAINING-BASED MORL APPROACHES?

We first compare the orchestration methods against training-based MORL baselines on both applications, using episode length, Pareto frontier quality (HV and sparsity), and objective-specific metrics as shown in Table 2.

---

[3] https://huggingface.co/meta-llama/Meta-Llama-3-8B

[4] https://huggingface.co/mistralai/Mistral-Small-24B-Instruct-2501

[5] https://www.anthropic.com/news/claude-3-7-sonnet

Table 2: Comparison of episode efficiency, Pareto front quality, and objective-specific metrics for two domains. **Bold** = domain- and method-wise best per metric. Shading = domain- and method-wise best overall per row (most bolded). SORL methods are not ranked.

| Method | Education | | | | | | MiniGrid | | | | | |
|---|---|---|---|---|---|---|---|---|---|---|---|---|
| | Len. ↓ | $R_{d_3}$ ↑ | $-R_{d_2}$ ↑ | $R_{d_1}$ ↑ | HV ↑ | Sparsity ↓ | Len. ↓ | Succ. ↑ | Gold ↑ | Lava ↓ | HV ↑ | Sparsity ↓ |
| SO-$d_1$ | 20.43 | 0.64 | 0.22 | 0.64 | 5.12 | 0.10 | 49.3 | 13 | 0 | 54 | 3.99 | 0 |
| SO-$d_2$ | 43.03 | 0.81 | 0.38 | 0.22 | 3.72 | 0.04 | 93.9 | 33.3 | 0 | 3 | 3.99 | 0 |
| SO-$d_3$ | 49.37 | 0.76 | 0.16 | 0.13 | 2.73 | 0.27 | 139.7 | 0 | 8 | 3 | 3 | 0 |
| MO-S | 47.43 | **0.79** | 0.15 | 0.20 | 3.39 | 0.10 | - | - | - | - | - | - |
| MO-RM | 48.20 | 0.64 | 0.30 | 0.14 | 3.47 | **0.05** | - | - | - | - | - | - |
| MO-MGDA | 43.07 | **0.79** | 0.36 | 0.23 | 3.85 | **0.05** | - | - | - | - | - | - |
| MO-pseudoS | - | - | - | - | - | - | 75.5 | 53 | 70 | **0** | 7.91 | 0.08 |
| KO-LLaMA | 38.63 | **0.75** | 0.15 | 0.24 | 3.40 | **0.08** | **16.9** | 28 | 9 | 67 | **7.68** | 0.18 |
| TO-LLaMA | 22.5 | 0.61 | 0.22 | 0.64 | **4.89** | 0.12 | 24.3 | 35 | 14 | 64 | 7.59 | **0.15** |
| RO-LLaMA | **21.1** | 0.65 | **0.24** | **0.65** | 4.5 | **0.08** | 25.4 | **42** | **15** | 47 | 7.66 | **0.15** |
| KO-Mistral | 40.2 | **0.75** | **0.34** | 0.25 | 3.81 | **0.05** | 19 | 17 | 11 | 74 | **7.91** | 0.25 |
| TO-Mistral | **21.47** | 0.73 | 0.28 | **0.48** | **5.18** | 0.10 | 19.3 | **27** | 14 | 64 | 6.46 | **0.05** |
| RO-Mistral | 21.73 | 0.71 | 0.33 | 0.40 | 5.11 | 0.09 | **11.6** | 20 | **47** | **51** | 7.90 | **0.05** |
| KO-Claude | 27.60 | **0.71** | 0.31 | 0.41 | 4.22 | **0.07** | **14.2** | 20 | 12 | 74 | 7.88 | 0.25 |
| TO-Claude | 24.53 | **0.71** | 0.32 | **0.50** | **4.77** | 0.11 | 32.6 | **60** | 18 | **27** | 7.78 | 0.11 |
| RO-Claude | **20.67** | **0.71** | **0.34** | 0.47 | 4.60 | 0.08 | 26.74 | 27 | **23** | 67 | **7.91** | **0.10** |

Table 3: Computational cost ($/run) for two domains, including training and inference costs. **Bold** = domain- and method-wise lowest total cost.

| Method | Education | | | MiniGrid | | |
|---|---|---|---|---|---|---|
| | Train ↓ | Infer ↓ | Total ↓ | Train ↓ | Infer ↓ | Total ↓ |
| KO-LLaMA | 0.97 | 0.00 | **0.97** | 0.97 | 0.00 | **0.97** |
| TO-LLaMA | 0.97 | 0.18 | 1.15 | 0.97 | 0.20 | 1.17 |
| RO-LLaMA | 0.97 | 0.42 | 1.39 | 0.97 | 0.51 | 1.48 |
| KO-Mistral | 0.97 | 0.00 | **0.97** | 0.97 | 0.00 | **0.97** |
| TO-Mistral | 0.97 | 0.17 | 1.14 | 0.97 | 0.16 | 1.13 |
| RO-Mistral | 0.97 | 0.44 | 1.41 | 0.97 | 0.24 | 1.21 |
| KO-Claude | 0.97 | 0.58 | **1.55** | 0.97 | 0.27 | **1.24** |
| TO-Claude | 0.97 | 0.71 | 1.68 | 0.97 | 2.47 | 3.44 |
| RO-Claude | 0.97 | 0.89 | 1.86 | 0.97 | 2.23 | 3.20 |

Across both domains, SO-$d_i$ policies predictably excel on their corresponding objective-specific metrics but fail to balance trade-offs, highlighting the MO nature of the tasks. MORL baselines, by contrast, achieve better overall trade-offs, confirming their effectiveness as upper-bound references.

Orchestration methods consistently achieve lower episode lengths than MORL baselines, indicating greater decision-time efficiency. In the Education domain, the best orchestrator in each LLM backbone (shaded) achieves higher HV and comparable sparsity to MO-MGDA, demonstrating competitive Pareto frontier quality. In MiniGrid, orchestrators attain HV scores close to MO-pseudoS, again with similar sparsity, despite lacking access to MO supervision during training.

On per-objective metrics, orchestrators in the Education domain often reach non-dominated performance compared to MO-MGDA, confirming their capacity to adaptively trade off objectives. In MiniGrid, however, the best orchestrator in each LLM backbone (shaded) underperforms MO-pseudoS in per-objective metrics. This suggests that in more complex environments, orchestration alone cannot fully replicate the benefits of joint multi-objective training. For instance, RO-Claude achieves a success rate of 27%, gold collection of 23%, and lava collision of 67%, whereas MO-pseudoS reaches 53%, 70%, and 0%, respectively. The performance gap stems from the fact that each SO policy was trained in a simplified environment tailored to a single objective.

Second, we report the total computational cost ($/run) of orchestrators in Table 3. We assume training a new MO policy costs $9.7/run, while each SO policy costs only $0.97/run and can be reused across objectives. Orchestrator's inference cost includes LLM token pricing, tool invocation ($0.002/run). Orchestrators avoid the $9.7/run cost of retraining MO policies by reusing existing SO policies ($0.97/run). Their total cost stays below one-third of retraining an MO policy across all applications and orchestrators. RO-Claude is the most expensive due to both higher token pricing and frequent tool calls. MiniGrid tasks incur higher inference costs than Education due to larger state/action spaces, particularly under Claude. Details and pricing assumptions are provided in Appendix A.6.

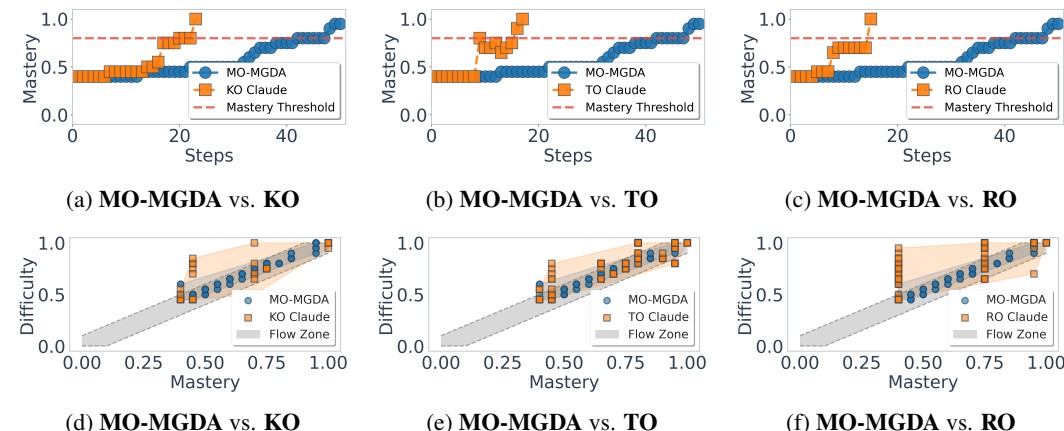

(a) **MO-MGDA** vs. **KO**    (b) **MO-MGDA** vs. **TO**    (c) **MO-MGDA** vs. **RO**

(d) **MO-MGDA** vs. **KO**    (e) **MO-MGDA** vs. **TO**    (f) **MO-MGDA** vs. **RO**

Figure 4: Mastery progression and flow alignment of **MO-MGDA** vs. **KO**, **TO**, and **RO**, with initial mastery level of 0.4.

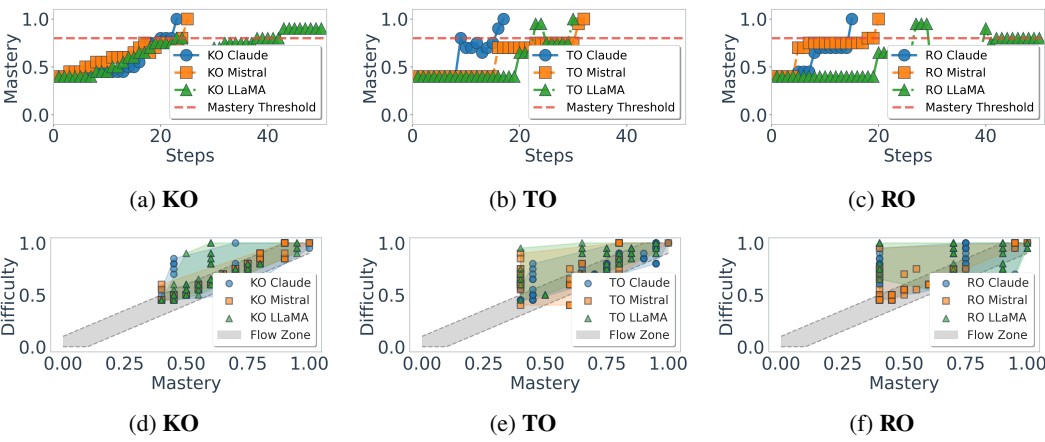

(a) **KO**    (b) **TO**    (c) **RO**

(d) **KO**    (e) **TO**    (f) **RO**

Figure 5: Mastery progression and flow alignment of **KO**, **TO**, **RO** with various LLM backbones.

Finally, Fig. 4 compares the *mastery progression* (panels 4a–4c) and *flow alignment* (panels 4d–4f) between MO-MGDA and orchestration methods on the Education application.

MO-MGDA demonstrates smoother mastery progression over time, reflecting gradual skill improvement driven by Bellman-consistent RL updates. It generally takes more steps to reach the mastery threshold due to conservative exploration. In contrast, all orchestrators (KO/TO/RO) reach mastery with fewer steps and display a steep late-phase rise in mastery. It indicates aptitude-driven question selection that accelerates learning once sufficient foundation is built. However, their mastery progression curves also show larger variance, reflecting the less stable nature of heuristic reasoning compared to end-to-end RL training. For example, in panel 4b, **TO** exhibits oscillations in mastery levels around step 13.

Flow alignment plots show that MO-MGDA consistently recommends questions within the flow zone, indicating stable difficulty alignment with student ability. This cautious behavior supports consistent learning but may slow mastery gains. Orchestrators, by contrast, select questions that span both the flow zone and above-flow regions, demonstrating a broader difficulty range. This behavior helps exploit learning opportunities just beyond the current mastery level (i.e., aptitude-based exploration). Overall, orchestration achieves comparable mastery outcomes with more adaptive, albeit less stable, learning paths.

### 4.2.2 **RQ2:** WHICH CONTEXT COMPONENTS MOST STRONGLY CONTRIBUTE TO THE ORCHESTRATION CAPABILITIES?

**Effect of LLM backbones.** We compare orchestration performance across LLaMA, Mistral, and Claude to assess how well different backbones utilize context. As shown in Table 2, LLaMA un-

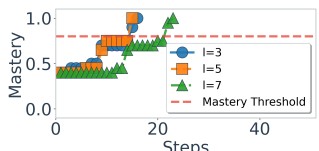

Figure 6: Impact of varying $\ell$ on **RO**'s mastery progression performance.

Table 4: Human-centered evaluation.

| | Mental | Temporal | Performance | Effort | Frustration |
|---|---|---|---|---|---|
| **KO** | 3 | 3 | 5 | 4 | 2 |
| **TO** | 4 | 3 | 4 | 4 | 3 |
| **RO** | 3 | 2 | 4 | 3 | 2 |

Table 5: Ablation study on memory components on the education domain. We compare each orchestrator vs. removing $c_{\text{mem-s}}$, $c_{\text{mem-l}}$, or both. Scalarized reward change, $\Delta_O$, indicates improvement over the respective orchestrator. **Bold** = the least degradation (i.e., best performance) within each orchestrator group.

| Method | LLaMA 3 8B | | | | | Mistral 24B | | | | | Claude 3.7 Sonnet | | | | |
|---|---|---|---|---|---|---|---|---|---|---|---|---|---|---|---|
| | $R_{d_3}$ | $R_{d_2}$ | $R_{d_1}$ | $\Delta_O$ | Cost | $R_{d_3}$ | $R_{d_2}$ | $R_{d_1}$ | $\Delta_O$ | Cost | $R_{d_3}$ | $R_{d_2}$ | $R_{d_1}$ | $\Delta_O$ | Cost |
| **KO** | 0.58 | 0.08 | 0.66 | – | $0.0048 | 0.88 | 0.68 | 0.25 | – | $0.0037 | 0.84 | 0.56 | 0.26 | – | $1.14 |
| $- c_{\text{mem-s}}$ | 0.85 | 0.55 | 0.35 | +0.14 | 0.0043 | 0.88 | 0.68 | 0.25 | **+0.00** | 0.0035 | 0.88 | 0.65 | 0.23 | +0.03 | 1.05 |
| $- c_{\text{mem-l}}$ | 0.82 | 0.67 | 0.36 | **+0.17** | 0.0038 | 0.88 | 0.65 | 0.23 | -0.01 | 0.0076 | 0.85 | 0.66 | 0.30 | **+0.05** | 1.077 |
| – both | 0.77 | 0.65 | 0.35 | +0.15 | 0.0032 | 0.88 | 0.65 | 0.17 | -0.03 | 0.0061 | 0.86 | 0.44 | 0.19 | -0.05 | 0.99 |
| **TO** | 0.79 | 0.15 | 0.25 | – | $0.0046 | 0.67 | 0.20 | 0.51 | – | $0.012 | 0.62 | 0.40 | 0.37 | – | $2.96 |
| $- c_{\text{mem-s}}$ | 0.59 | 0.08 | 0.67 | +0.05 | 0.0042 | 0.73 | 0.15 | 0.49 | **+0.00** | 0.0095 | 0.83 | 0.43 | 0.39 | +0.09 | 2.60 |
| $- c_{\text{mem-l}}$ | 0.75 | 0.28 | 0.47 | **+0.10** | 0.0047 | 0.91 | 0.15 | 0.18 | -0.05 | 0.0086 | 0.77 | 0.49 | 0.53 | **+0.14** | 3.01 |
| – both | 0.69 | 0.13 | 0.53 | +0.05 | 0.0057 | 0.76 | 0.20 | 0.31 | -0.03 | 0.0073 | 0.73 | 0.13 | 0.58 | +0.02 | 2.74 |
| **RO** | / | / | / | / | / | 0.80 | 0.39 | 0.40 | – | $0.0091 | 0.80 | 0.47 | 0.28 | – | $2.45 |
| $- c_{\text{mem-s}}$ | / | / | / | / | / | 0.74 | 0.38 | 0.63 | **+0.05** | 0.0080 | 0.83 | 0.50 | 0.50 | **+0.08** | 2.16 |
| $- c_{\text{mem-l}}$ | / | / | / | / | / | 0.74 | 0.17 | 0.37 | -0.10 | 0.0089 | 0.78 | 0.48 | 0.36 | +0.02 | 2.45 |
| – both | / | / | / | / | / | 0.77 | 0.25 | 0.49 | -0.02 | 0.0078 | 0.71 | 0.31 | 0.55 | +0.00 | 2.49 |

derperforms across all metrics for two domains, i.e., requiring longer episodes, achieving lower HV, and producing unstable mastery gains (Fig. 5). From the flow alignment plots, LLaMA-based orchestrators also often recommend overly difficult questions early on, reflecting weak reasoning over policy constraints and student state. In contrast, Mistral- and Claude-based orchestrators yield more stable mastery progression and better flow alignment, suggesting stronger understanding of instruction, memory, and policy descriptions. This indicates that larger LLMs more effectively leverage structured context components, leading to better orchestration decisions.

**Effect of policy context designs.** We analyze the impact of context design by comparing KO, TO, and RO within each LLM backbone in Table 2. For two applications, across all backbones, RO consistently achieves the most shaded rows, indicating superior performance. TO is second-best, while KO performs worst.

This trend highlights key differences in context components. KO relies on static policy descriptions ($c_{\text{know}}$), which lack adaptivity to changing states. Even with behavioral hints and examples, they fail to support context-sensitive reasoning. In contrast, TO and RO include $c_{\text{tools}}$, enabling real-time policy queries. RO further adds $c_{\text{reflect}}$ to allow foresight into long-term outcomes. These components make RO more context-aware and lead to stronger orchestration performance.

We further analyze RO's internal design by varying its look-ahead horizon $\ell$ using the education application (Fig. 6). We observe that moderate values ($\ell = 3$ or $\ell = 5$) lead to smooth and rapid mastery progression, while overly long horizons ($\ell = 7$) introduce instability and delay. This suggests that limited foresight is sufficient for effective reasoning, whereas excessive reflection may overwhelm the LLM's context capacity.

**Effect of $c_{\text{mem}}$.** We investigate the effect of memory components using the education application. As shown in Table 5, removing memory components from orchestrators reveals several patterns. First, short- and long-term memory are not always required for strong per-objective performance. For example, LLaMA performs significantly better without memory, suggesting that smaller models with limited reasoning ability may be confused by additional memory rather than aided by it. In contrast, larger models benefit more from memory, such as Mistral, where removal leads to notable reward decrease. Third, removing memory does not substantially reduce cost because LLM inference cost depends on input and output tokens; without memory, the orchestrator may need to re-derive prior context by longer reasoning.

### 4.2.3 QUALITATIVE ANALYSIS WITH LLM-JUDGE

Taking the education application as an example, we conduct two qualitative analyses using GPT-5[6] as the LLM-judge (Gu et al., 2024): (1) orchestrator trajectory comparison; and (2) human-centric assessment of experiential dimensions.

**Qualitative analysis of orchestrator trajectories.** We asked an LLM-judge to examine the logs of recommended trajectories and evaluate the following metrics:

- *Adaptivity Across Turns.* Measures the orchestrator's ability to update and personalize recommendations in response to ongoing learner feedback or simulated responses;
- *Progression and Scaffolding:* Evaluates the logical progression and scaffolding in the sequence of recommendations, ensuring smooth and supportive advancement of learner skill;
- *Exploration vs. Exploitation:* Quantifies the balance between trying new policies (exploration) and repeating known effective ones (exploitation), reflecting strategic decision-making;
- *Policy Diversity:* Captures the variety and distinctiveness in the sequence of selected policies, indicating whether the orchestrator explores different instructional strategies.

The detailed results are shown in Appendix A.7. In summary, we find that **TO** and **RO** consistently achieve the highest scores across most trajectory-based metrics, while **KO**'s limited adaptivity leads to lower scores in adaptivity, exploration, and diversity. These results demonstrate the benefits of reflection (single or parallel) in delivering robust and pedagogically effective decisions, especially in dynamic or personalized learning scenarios.

**Human-centric analysis.** To complement our previous analysis, we propose to adapt the NASA-TLX metrics [7] to evaluate human-like experiential dimensions, as follows:

- *Mental Demand*: cognitive workload required to interact with or understand recommendations;
- *Temporal Demand*: perceived learning pace and time to completion;
- *Performance*: how well the orchestrator helps the learner attain mastery;
- *Effort* and *Frustration*: subjective cost and negative emotional states induced by orchestration.

As shown in Table 4, **KO** yields low frustration and high performance, suggesting it recommends easier questions. However, slower mastery gains from easy items lead to higher overall effort to attain mastery. **TO** induces more mental load and frustration, with lower performance, implying a bit harder questions. **RO** minimizes effort and temporal demand, offering the most streamlined user experience. These results underscore the importance of balancing decision quality and learner experience in orchestrator design.

## 5 CONCLUSION

We proposed a context-driven, zero-shot orchestration framework using LLMs for multi-objective decision making at inference time. Experiments across two domains (education and control) show that our orchestrators can match or exceed training-based MORL baselines in decision quality, while reducing computational cost by over $3\times$.

While our framework supports MO-SDM through context encoding, we do not yet explicitly model user preferences or generate diverse trade-off solutions. Future work will extend our method to preference-conditioned orchestration and multi-solution generation. In addition, we currently assume deterministic transitions; generalizing to stochastic or partially observable settings remains an open challenge. Finally, we aim to explore long-horizon agent self-improvement and memory consolidation to enhance adaptability in evolving environments.

---

[6]https://openai.com/gpt-5/
[7]https://en.wikipedia.org/wiki/NASA-TLX

**Ethic Statement.** This work uses only publicly available data and does not involve human subjects. All authors adhere to the ICLR Code of Ethics.

**Reproducibility Statement.** We provide full implementation details in the paper and appendix, including hyperparameters, datasets, and prompt templates. All code and scripts are available in the anonymous supplementary material to facilitate replication.

**Use of LLMs.** This paper studies LLM-based orchestrators and thus extensively employs LLMs (LLaMA-3-8B, Mistral-24B, and Claude-3.7-Sonnet) as research subjects. The use of LLMs is explicitly reported in Section 4. In addition, GPT-5 was used as an LLM-judge in Section 4.2.3 to qualitatively evaluate orchestrator trajectories and experiential dimensions. No LLM was used for paper writing or enhancing.

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

# A APPENDIX

## A.1 RELATED WORK

Reinforcement learning (RL) has achieved remarkable success in *multi-objective sequential decision-making* (MO-SDM) tasks, such as Go (Silver et al., 2017), Atari (Mnih et al., 2015), and Dota-2 (Berner et al., 2019), with performance comparable or better than human-beings. Most of these breakthroughs, however, fall under single-objective RL (SORL), where the agent optimizes a single scalar reward. While many sequential decision-making problems can be reduced to SORL, a growing number of real-world tasks naturally involve multiple and conflicting objectives, for instance, increasing performance, reducing gap, and keeping aptitude in education question recommendation (Coello, 2000; Abels et al., 2019; Van Moffaert & Nowé, 2014; Hayes et al., 2021). Multi-Objective RL (MORL) extends RL to such settings, aiming to learn policies that achieve desirable trade-offs between objectives. MORL algorithms fall into three categories: (1) scalarization (Roijers et al., 2013), (2) Pareto-based (Roijers et al., 2013; Van Moffaert & Nowé, 2014), (3) gradient-based (Yu et al., 2020), and more (Abels et al., 2019; Pirotta et al., 2015; Deb et al., 2002; Vamplew et al., 2011; Rădulescu et al., 2020). Recent methods (Alegre et al., 2023; Röpke et al., 2024; Lu et al., 2023) have advanced sample efficiency and theoretical guarantees on convexity and Pareto optimality.

However, these approaches assume that the full set of objectives is known *a priori* during training, which is rarely the case in practice (Hayes et al., 2021). Moreover, objectives often vary across contexts. Scalarized or fixed-policy methods must then be redesigned and retrained (Yu et al., 2020; Van Moffaert & Nowé, 2014; Roijers et al., 2013), which is prohibitively expensive. Consequently, these limitations make traditional MORL methods ill-suited for deployment scenarios where objective configurations are uncertain or may shift at inference time.

In this work, we take a complementary perspective and ask whether a set of pre-trained SO policies can be orchestrated at inference time to approximate Pareto-efficient behaviors *without retraining*. A related line of work explores policy reuse and modularization in RL. Fernández & Veloso (2006) first proposed *probabilistic policy reuse*, where an agent accelerates learning by probabilistically selecting among exploiting its current policy, exploring new actions, or leveraging a stored prior policy. Subsequent work introduced high-level policy sketches to describe sequences of subtasks and compose modular sub-policies across tasks (Andreas et al., 2017). Later, meta-RL approaches emerged, where a shared policy network is trained over a distribution of tasks, and a learned task embedding captures task-specific variations, enabling fast adaptation to unseen but related tasks (Lan et al., 2019). Unlike these training-time reuse approaches, which rely on gradient-based adaptation, our method aim to achieve inference-time MO-SDM. More recently, Nikookar et al. (2025) proposed a graph-based model to achieve MO optimization by reusing pre-trained RL policies across tasks without additional training. This line of work is conceptually closer to ours, as it also focuses on post-training policy reuse; however, it is restricted to cases where the new task's reward is a linear combination of those of pre-trained policies. In contrast, our approach generalizes to conflicting objectives.

Recent advances in large language models (LLMs) have shown that they can serve as powerful decision-making agents capable of reasoning, reflection, and tool use (Shinn et al., 2023; Zhou et al., 2023; Yao et al., 2022). Context engineering (Mei et al., 2025) further formalizes how structured prompts, comprising knowledge, memory, feedback, and more components, shape LLM reasoning. In contrast to recent work where RL is applied to optimize LLMs, such as reinforcement learning from human feedback (RLHF) (Bai et al., 2022) and MO alignment for language models (Wang et al., 2024; He & Maghsudi, 2025; Jafari et al., 2024), our goal is to explore a distinct question: *Can we solve MO-SDM problems by orchestrating existing SO policies using LLMs at inference time, without learning a new MO policy from scratch?*

## A.2 FULL PROMPT OF CONTEXT COMPONENTS FOR QUESTION RECOMMENDATION

In Sec. 3, we describe each context component. Here, we provide the full prompt of each context components for question recommendation in the education application. We first present the system and user prompts of KO. The system prompt (Figure 7) includes $c_{\text{instr}}$ and $c_{\text{mem-l}}$, while the user prompt (Figure 8) consists of $c_{\text{query}}$, $c_{\text{know}}$, $c_{\text{state}}$, and $c_{\text{mem-s}}$. We then provide examples to illustrate

instantiations of $c_{\text{mem-l}}$ (Figure 10), $c_{\text{mem-s}}$ (Figure 11), $c_{\text{know}}$ (Figure 12) and $c_{\text{tool}}$ (Figure 13). For long-term memory $c_{\text{mem-l}}$, we prompted the LLM to iteratively summarize past interactions, and the prompt is in Figure 9. Finally, we include the system and user prompts of TO (Figures 14- 15) and RO (Figures 16- 17).

```
System Message:
You are an expert educational recommendation orchestrator.
Your role is to select exactly one pre-trained policy which can then
↪   recommend the next set of questions about the target skill bundle
↪   for a student. The target skill bundle is: {target_skill_bundle}.

Each pre-trained policy is a fixed function that maps a student state
↪   to an action.
- Input: a student state, including mastery levels, accuracy per skill
↪   × difficulty, and failed question summaries.
- Output: one of three possible action types, each recommending 5
↪   questions:
  - Action 0: Failed Questions | Recommend previously failed or closely
   ↪   related questions to review mistakes and close knowledge gaps.
  - Action 1: Easy Questions | Recommend familiar, low-difficulty
   ↪   questions that the student is very likely to answer correctly to
   ↪   build performance and confidence.
  - Action 2: High-Aptitude Questions | Recommend the most challenging
   ↪   questions (difficulty > mastery), prioritizing the largest
   ↪   difficulty{mastery gap to maximize challenge.

Policies are static decision functions with fixed optimization
↪   objectives and performance statistics. They cannot change their
↪   behavior during orchestration.

## Objectives
{objectives}

## Guardrails
- Always select exactly one policy.
- Always provide a clear reasoning before selection.
- Always output in strict JSON format: {format_instructions}

## Instructions
- Base your decision on the provided context (student state, policy
↪   information, recent interactions, and long-term summary of past
↪   interactions).
- When objectives conflict, trade off carefully and prefer consistent
↪   high-performing policies.
- Adapt your choice based on recent history of student learning
↪   progression.

c_mem-l
```

Figure 7: System Prompt of **KO**, including $c_{\text{instr}}$ and $c_{\text{mem-l}}$

```
User Message:
## Query
Select one best policy to use for recommending the next set of
↪  questions to the student, based on the current student state and
↪  the available pre-trained policies.

c_know

## Student State:
- Skill Mastery Levels: {mastery}
- Number of Failed Questions: {number_of_failed_questions}

c_mem-s
```

Figure 8: User Prompt of **KO**, including $c_{query}$, $c_{know}$, $c_{state}$, and $c_{mem-s}$

```
You are an educational AI assistant that summarizes student learning
↪  interactions.
Your task is to create a concise summary of the student's learning
↪  progress and the orchestrator's policy selection patterns.

Current summary:
{summary}

New interactions:
{new interactions}

Create a new summary that captures:
1. Student's learning patterns (mastery levels, performance trends)
2. Orchestrator's policy selection patterns (which policies work well
↪  for different student states)
3. Key insights about what strategies lead to successful learning
↪  outcomes

New summary:
```

Figure 9: Prompt template used for generating a long-term memory component $c_{mem-l}$.

```
# Learning Interaction Summary

## Student Learning Patterns
- Current mastery level is at 0.4 (40%)
- Student has failed 2 questions so far
- Recent performance appears positive with a performance reward of 0.80

## Orchestrator Policy Patterns
- Policy "ppo_2" was selected for the current student state
- This policy recommended easy questions as the action
- The decision yielded high performance reward (0.80), no gap reward,
↪  and minimal aptitude reward (0.08)

## Key Insights
- Recommending easy questions appears to be an effective strategy for
↪  this student at their current mastery level (0.4)
- The orchestrator is prioritizing performance improvement over closing
↪  knowledge gaps at this stage
- Given the failed questions history, the system is likely taking a
↪  more conservative approach to build confidence before increasing
↪  difficulty
```

Figure 10: An example of long-term memory components $c_{\text{mem-l}}$ for **KO**. This component is a part of system prompt.

```
## Recent Orchestrator Interactions (Most recent last)
1. Student State:
- Mastery: [0.4]
- Number of Failed Questions: 2
   Orchestrator Selected Policy: ppo_2
Policy Action: recommend easy questions
Student Outcome:
- Correctness: Mostly correct
- Mastery Change: Stable
- Reward: performance=0.80, gap=0.00, aptitude=0.08
```

Figure 11: A example of short-memory components $c_{\text{mem-s}}$ for **KO**. This component is a part of user prompt.

```
## Available Policies
{
  "policies": [
    {
      "name": "ppo_0",
      "optimized_objective": "aptitude",
      "overall_strength": "high",
      "stability": "volatile",
      "behavior_hint": "tends to choose Action2 (Aptitude) in any
      ↪  cases",
      "applicability": "Ideal when mastery is high; accelerates growth
      ↪  by challenging with hardest items.",
      "failure_modes": "May lead to stagnation and frustration in early
      ↪  stages of learning.",
      "example_snippets": [
        "above_threshold + no_fail \u2192 Action2 (Aptitude) \u2192
        ↪  aptitude_high, perf_low"
      ],
      "avg_scalar_reward": 11.782746666666668,
      "std_scalar_reward": 7.164085966529313,
      "objectives": [
        "aptitude"
      ]
    },
    ...
  ],
  "rubric": {
    "accuracy": {
      "high": "greater than or equal to 0.8",
      "medium": "in between 0.6 and 0.79",
      "low": "below 0.6"
    },
    "mastery": {
      "above threshold": "greater than or equal to 0.8",
      "below threshold": "below 0.8"
    },
    "reward": {
      "high": "normalized reward greater than or equal to 0.6",
      "medium": "normalized reward in between 0.4 and 0.6",
      "low": "normalized reward below 0.4"
    },
    "stability": {
      "stable": "std below 0.05",
      "moderate": "std in between 0.05 and 0.15",
      "volatile": "std greater than or equal to 0.15"
    }
  }
}
```

Figure 12: An example of the knowledge component $c_{\text{know}}$ for **KO**. This component is a part of user prompt.

```
## Available Tools
{
  "schema": {
    "input": {
      "student state": {
        "mastery": "Dictionary of skill names to float values between 0
        ↪  and 1. Example: {'Linear Algebra': 0.75}",
        "accuracy": "Dictionary of (skill, difficulty) to float
        ↪  accuracy values. Example: {('Linear Algebra', 'easy'):
        ↪  0.9}",
        "failures": "Dictionary of skill \u00d7 difficulty to failed
        ↪  question ratio (0-1). Example: {('Linear Algebra', 'easy'):
        ↪  0.2}"
      }
    },
    "output": {
      "action": "String. One of ['Failed Questions', 'Easy Questions',
      ↪  'High-Aptitude Questions']",
      "recommended questions": "List of tuples. Each is a question
      ↪  identifier. Example: [('Q001', 'easy'), ('Q015', 'medium'),
      ↪  ('Q103', 'hard'), ('Q042', 'easy'), ('Q087', 'medium')]",
      "student feedback": {
        "performance reward": "Float. Normalized to [0, 1]",
        "gap reward": "Float. Normalized to [0, 1]",
        "aptitude reward": "Float. Normalized to [0, 1]",
        "mastery change": "String. One of ['improved', 'declined',
        ↪  'unchanged']"
      }
    }
  },
  "tools": [
    {
      "name": "ppo_0",
      "description": "tends to choose Action2 (Aptitude) in any cases",
      "optimized_objective": "aptitude"
    },
    ...
  ]
}
```

Figure 13: An example of the tool component $c_{\text{tool}}$ for **TO**. This component is a part of user prompt.

```
System Message:
You are an expert educational recommendation orchestrator.
Your role is to select exactly one pre-trained policy which can then
↪  recommend the next set of questions about the target skill bundle
↪  for a student. The target skill bundle is: {target_skill_bundle}.

Each pre-trained policy is a fixed function that maps a student state
↪  to an action.
- Input: a student state, including mastery levels, accuracy per skill
↪  × difficulty, and failed question summaries.
- Output: one of three possible action types, each recommending 5
↪  questions:
  - Action 0: Failed Questions | Recommend previously failed or closely
  ↪  related questions to review mistakes and close knowledge gaps.
  - Action 1: Easy Questions | Recommend familiar, low-difficulty
  ↪  questions that the student is very likely to answer correctly to
  ↪  build performance and confidence.
  - Action 2: High-Aptitude Questions | Recommend the most challenging
  ↪  questions (difficulty > mastery), prioritizing the largest
  ↪  difficulty{mastery gap to maximize challenge.

Policies are static decision functions with fixed optimization
↪  objectives and performance statistics. They cannot change their
↪  behavior during orchestration.

## Objectives
{objectives}

## Guardrails
- Always select exactly one policy as either a tool call to get student
↪  feedback or final decision.
- Avoid calling the same policy multiple times during tool calls
↪  because it will not be able to provide new information.
- Always provide a clear reasoning.
- You have at maximum {max_tool_calls} tool calls to get student
↪  feedback of the tools.
- Always make a final decision once you have reached the maximum number
↪  of tool calls.
- Always output in strict JSON format: {format_instructions}

## Instructions
- Base your decision on the provided context (student state, available
↪  tools, tool call history, recent interactions, and long-term
↪  summary of past interactions).
- When objectives conflict, trade off carefully and prefer consistent
↪  high-performing policies.
- Adapt your choice based on the the student feedback (rewards per
↪  objective, recommended questions' difficulty level and mastery
↪  changes) from the tool calls.
- Once you get enough information from the tool calls, make a final
↪  decision about which pre-trained policy to use for question
↪  recommendation.

$c_{\text{mem-l}}$
```

Figure 14: System Prompt of **TO**, including $c_{\text{instr}}$ and $c_{\text{mem-l}}$

```
User Message:
## Query
Select one best policy to use for recommending the next set of
↪   questions to the student, based on the current student state, the
↪   available tools, and the tool call history.

c_tool

## Student State:
- Skill Mastery Levels: {mastery}
- Number of Failed Questions: {number_of_failed_questions}

c_mem-s
```

Figure 15: User Prompt of **KO**, including $c_{query}$, $c_{tool}$, $c_{state}$, and $c_{mem-s}$

```
System Message:
You are an expert educational recommendation orchestrator.
Your role is to select exactly one pre-trained policy which can then
↪  recommend the next set of questions about the target skill bundle
↪  for a student. The target skill bundle is: {target_skill_bundle}.

Each pre-trained policy is a fixed function that maps a student state
↪  to an action.
- Input: a student state, including mastery levels, accuracy per skill
↪  × difficulty, and failed question summaries.
- Output: one of three possible action types, each recommending 5
↪  questions:
  - Action 0: Failed Questions | Recommend previously failed or closely
  ↪  related questions to review mistakes and close knowledge gaps.
  - Action 1: Easy Questions | Recommend familiar, low-difficulty
  ↪  questions that the student is very likely to answer correctly to
  ↪  build performance and confidence.
  - Action 2: High-Aptitude Questions | Recommend the most challenging
  ↪  questions (difficulty > mastery), prioritizing the largest
  ↪  difficulty{mastery gap to maximize challenge.

Policies are static decision functions with fixed optimization
↪  objectives and performance statistics. They cannot change their
↪  behavior during orchestration.

## Objectives
{objectives}

## Guardrails
- Always select exactly one policy as either a tool call to get student
↪  feedback after {rollout_steps}-step rollout or final decision.
- Avoid calling the same policy multiple times during tool calls
↪  because it will not be able to provide new information.
- Always provide a clear reasoning.
- You have at maximum {max_rollouts} tool calls to get student feedback
↪  of the tools.
- Always make a final decision once you have reached the maximum number
↪  of tool calls.
- Always output in strict JSON format: {format_instructions}

## Instructions
- Base your decision on the provided context (student state, available
↪  tools, rollout history, recent interactions, and long-term summary
↪  of past interactions).
- When objectives conflict, trade off carefully and prefer consistent
↪  high-performing policies.
- Adapt your choice based on the the student feedback (rewards per
↪  objective, recommended questions' difficulty level and mastery
↪  changes) from the tool calls.
- Once you get enough information from the tool calls, make a final
↪  decision about which pre-trained policy to use for question
↪  recommendation.

c_{mem-l}
```

Figure 16: System Prompt of **RO**, including $c_{instr}$ and $c_{mem-l}$

```
User Message:
## Query
Select one best policy to use for recommending the next set of
↪   questions to the student, based on the current student state, the
↪   available tools, and the tool call rollout history.

c_tool

## Student State:
- Skill Mastery Levels: {mastery}
- Number of Failed Questions: {number_of_failed_questions}

c_reflect

c_mem-s
```

Figure 17: User Prompt of **RO**, including $c_{\text{query}}$, $c_{\text{tool}}$, $c_{\text{state}}$, $c_{\text{reflect}}$, and $c_{\text{mem-s}}$

### A.3 TRAINING RL POLICIES

We implement all SO and MO policies using Proximal Policy Optimization (PPO) (Schulman et al., 2017) from Stable-Baselines3 (Raffin et al., 2021), given its superior sample efficiency and stability across both discrete and continuous control tasks. PPO is a policy gradient algorithm designed to improve stability by constraining policy updates (Schulman et al., 2017). Instead of directly maximizing the standard policy gradient objective, PPO introduces a clipped surrogate objective:

$$L^{\text{CLIP}}(\theta) = \mathbb{E}_t \big[ \min \big( r_t(\theta) A_t, \, \text{clip}(r_t(\theta), 1 - \epsilon, 1 + \epsilon) A_t \big) \big], \tag{1}$$

where $r_t(\theta) = \frac{\pi_\theta(a_t|s_t)}{\pi_{\theta_{\text{old}}}(a_t|s_t)}$ is the probability ratio, $A_t$ is the advantage, and $\epsilon$ is the clipping threshold.

To train policies under multiple objectives, we consider the following three formulations:

1. **Scalarization (MO-S):** A fixed-weight linear combination of individual objective rewards:

$$r_{\text{scalar}}(s, a, s') = \sum_{d_i} w_{d_i} \cdot r_{d_i}(s, a, s'), \tag{2}$$

where $w_o = 1/|\mathcal{D}|$ (uniform) unless otherwise specified.

2. **Reward Machine (MO-RM):** A finite-state automaton with two reward phases:
   - $u_0$ (*Performance phase*): $r = 0.5 \cdot r_{\text{perf}} + 0.5 \cdot r_{\text{gap}}$
   - $u_1$ (*Aptitude phase*): $r = 0.5 \cdot r_{\text{gap}} + 0.5 \cdot r_{\text{apt}}$

   The RM transitions between $u_0$ and $u_1$ based on average mastery: if avg_mastery $\geq 0.6$, transition to $u_1$; otherwise revert to $u_0$.

3. **Gradient-based (MO-MGDA):** We extend PPO to the multi-objective setting by applying the Multiple Gradient Descent Algorithm (MGDA) (Désidéri, 2009). At each update step, we compute the surrogate loss $L_i^{\text{CLIP}}(\theta)$ for each objective $i$, and optimize a convex combination of their gradients:

$$\min_{\alpha \in \Delta^k} \left\| \sum_{i=1}^{k} \alpha_i \nabla_\theta L_i^{\text{CLIP}}(\theta) \right\|_2^2, \tag{3}$$

where $\Delta^k$ is the $k$-simplex, and $L_i^{\text{CLIP}}(\theta)$ is the PPO surrogate loss in Equation 1 under the $i$-th objective reward signal. This ensures a common descent direction that balances all objectives.

### A.4 QUESTION RECOMMENDATION MDP FORMULATION

We formulate the student learning environment as a Markov Decision Process (MDP) $\mathcal{M} = \langle \mathcal{S}, \mathcal{A}, \{\mathcal{R}_i\}_{i=1}^m, P, H \rangle$, where:

- **States $\mathcal{S}$.** Each state $s \in \mathcal{S}$ captures the student's mastery vector across skills, and per-skill-and-difficulty accuracy and failure statistics.

- **Actions** $\mathcal{A}$**.** Each action $a \in \mathcal{A}$ corresponds to a recommendation strategy that outputs a set of questions $\mathcal{Q}_a$:

$$a_{\text{easy}} = \{q \mid d(q) > m(s),\ q \sim q_{\text{success}}\},$$
$$a_{\text{failed}} = \{q \in \mathcal{Q}_{\text{fail}} \mid |d(q) - m(s)| < \delta\},$$
$$a_{\text{chall}} = \{q \in |d(q) - m(s)| > \delta\},$$

  where $d(q)$ is question difficulty, $m(s)$ is current mastery of the student, and $\delta = 0.2$. Similarity $q \sim q_{\text{success}}$ is measured by Euclidean distance over difficulty levels (Bouarour et al., 2024).

- **Transition** $P$**.** Defines mastery updates and student responses. Details are provided below.

- **Rewards** $\{\mathcal{R}_i\}$**.** Each learning objective defines a reward function (performance, gap, aptitude).

  1. **Aptitude:** encourage long-term growth by rewarding challenging questions:

$$\mathcal{R}_{\text{apt}}(s, a) = \frac{1}{|\mathcal{Q}_a|} \sum_{q \in \mathcal{Q}_a} \max\left\{0, \frac{\text{difficulty}(q) - \text{mastery}(q.\text{skill})}{1 - q.\text{skill}}\right\}.$$

  2. **Gap:** reward only when the student correctly answers a question that was previously answered incorrectly.

  3. **Performance:** reward correct answers, reinforcing accurate responses:

$$\mathcal{R}_{\text{perf}}(s, a) = \mathbb{I}[\text{correct answer}].$$

- **Horizon** $H$**.** We fix $H = 50$ steps.

**Mastery Updates.** We use the static mastery detection method N Consecutive Correct (NCC) (Kelly et al., 2016), where a skill is marked as mastered once the student answers $N$ consecutive questions correctly at their current mastery level of difficulty. We set $N = 2$. **Student Response Simulator - Item Response Theory (IRT) (Embretson & Reise, 2013).** The probability of giving a correct response given student ability $\theta_s$ and task difficulty $d$ is

$$P(\text{correct} \mid \theta_s, d) = \frac{1}{1 + \exp(-a(\theta_s - d))},$$

with discrimination parameter $a$. We set $a = 4, \theta_s = \text{avg\_mastery}$

**Implementation Details.** Each episode begins with a five-question warm-up phase to initialize mastery estimates. At each step, the agent recommends $q = 5$ questions. Difficulty level is rescaled to 0-1, aligning to the mastery range. We set the target skill to be "linear algebra".

**Hyperparameters.** Table 6 summarizes the hyperparameters used in training SO policies using PPO and MO-MGDA policy for the education application. By default, we followed the recommended settings in Stable-Baselines3 (Raffin et al., 2021), with minor tuning for our environment.

Table 6: Education environment hyperparameters for SO policies and MO-MGDA policies.

| Parameter | SO policies | MO-MGDA |
|---|---|---|
| Total timesteps | 400,000 | |
| Hidden dimensions | (64, 64) | |
| $\gamma$ (discount factor) | 0.99 | |
| Learning rate | 1e-4 | 3e-3 |
| GAE lambda | 0.95 | 0.95 |
| Value coefficient | 0.5 | 0.5 |
| Entropy coefficient | 0.0 | 0.02 |
| Max grad norm | 0.5 | 0.5 |
| Target KL | 0.01 | 0.02 |
| Mini-batch size | 64 | 64 |
| Epochs per update | 10 | 10 |
| Steps per rollout | 256 | 2048 |
| Clip ratio | 0.1 | 0.2 |
| Normalize advantages | True | True |
| MGDA method | – | QP |

## A.5 MINIGRID MDP FORMULATION

We adopt the MiniGrid (Chevalier-Boisvert et al., 2023) to instantiate MO-SDM in a grid-world control domain. Each MiniGrid environment defines a Markov Decision Process (MDP) $\mathcal{M} = \langle \mathcal{S}, \mathcal{A}, \{\mathcal{R}_i\}, P, H \rangle$, with the following components:

- **States** $\mathcal{S}$**.** The state is a $7 \times 7 \times 3$ partial observable grid representing the agent's local field of view, encoded with object type, color, and state.
- **Actions** $\mathcal{A}$**.** The discrete action space contains 6 primitive actions: turn left/right, move forward, pick up object, drop object, and toggle door.
- **Transition** $P$**.** Transitions are deterministic given the current grid configuration and agent action. Episode ends upon reaching goal or timeout.
- **Reward Functions** $\{\mathcal{R}_i\}$**.** We define three SO tasks as follows:
  1. **Collect Gold:** Reward of $+1$ for picking up gold objects. Environment: `MiniGrid-GoToObjectPickup-6x6-N2-v0` (modified).
  2. **Avoid Lava:** Implicit reward via survival. Reward of $+1$ for reaching goal without stepping on lava. Environment: `MiniGrid-LavaGapS6-v0`.
  3. **Navigate:** Reward of $+1$ for reaching goal. Environment: `MiniGrid-Empty-6x6-v0`.
- **Horizon** $H$**.** We fix $H = 144$ steps.

**Multi-objective Environment.** In contrast to education environments that expose reward vectors, MiniGrid does not naturally provide multi-objective feedback during training. Following standard practice (Chevalier-Boisvert et al., 2023), we define a **MO-pseudoS** policy trained on a custom composite map (6x6 grid) that contains goal, lava, and gold items simultaneously. We assign a shaped scalar reward:

$$r_t = \mathbb{I}[\text{reach goal}] + \sum_{g \in \text{gold picked}} \mathbb{I}[g].$$

Avoiding lava yields no reward but is required for episode success. This setup simulates multiple objectives via scalarized feedback and serves as a pseudo-MO baseline to evaluate orchestration performance.

**Hyperparameters.** Table 7 summarizes the hyperparameters used in training SO and MO policies for MiniGrid. By default, we followed the recommended settings in Stable-Baselines3 (Raffin et al., 2021), with minor tuning for our environment.

Table 7: MiniGrid hyperparameters for SO policies, i.e., Nav ($d_1$), Lava ($d_2$), Gold ($d_3$) and the MO-pseudoS policy.

| Parameter | SO-$d_1$ | SO-$d_2$ | SO-$d_3$ | MO-pseudoS |
|---|---|---|---|---|
| Total timesteps | 200,000 | 200,000 | 200,000 | 50,000 |
| Policy | | | CnnPolicy | |
| Learning rate | 3e-4 | 3e-4 | 2e-4 | 3e-3 |
| $\gamma$ (discount factor) | | | 0.99 | |
| GAE lambda | | | 0.95 | |
| Clip range | | | 0.2 | |
| Value coefficient | | | 0.5 | |
| Entropy coefficient | 0.02 | 0.01 | 0.01 | 0.01 |
| Max grad norm | | | 0.5 | |
| Steps per rollout | | | 2048 | |
| Mini-batch size | | | 64 | |
| Epochs per update | | | 10 | |
| Features dimension | | | 128 | |

## A.6 COST AND ENERGY CONSUMPTION

### A.6.1 FORMULA

We compute both monetary cost and energy consumption per algorithm run. We take the education application as the example, each run involves $H = 50$ sequential decisions, and the computation differs for (1) LLM-based orchestrators and (2) MORL-based methods.

**Orchestrator Cost.** Each decision incurs both input and output token charges:

$$LLM\_cost\_per\_run = H \times (tokens_{in} \times price_{in} + tokens_{out} \times price_{out}),$$

$$LLM\_CO2e\_per\_run = H \times (tokens_{total} \times energy_{per\_token} \times grid_{kgCO2e/kWh}),$$

The values $tokens_{in}$ and $price_{in}$ are for input tokens, and the values $tokens_{out}$ and $price_{out}$ for output tokens. Based on USA average and commonly used assumptions[8], we consider $grid_{kgCO2e/kWh}$ to be $0.35$ kgCO2e per kWh. We also consider $energy_{per\_token}$ to be $1.4e-6$ kWh per token, based on (Jegham et al., 2025). The monetary cost of LLMs used in this work, including LLaMA 3 8B, Mistral 24B, and Claude 3.7 Sonnet, are summarized in Table 8.[9]

Table 8: LLM Pricing.

| LLM | Input (\$ / token) | Output (\$ / token) |
|---|---|---|
| Claude 3.7 Sonnet | 3.00E-06 | 1.50E-05 |
| Mistral 24B | 1.00E-08 | 1.50E-07 |
| LLaMA 3 8B | 3.00E-08 | 6.00E-08 |

**Cost of RL Methods.** Table 9 summarizes the assumptions used to estimate the cost and energy footprint of training and serving our RL policies. These values are grounded in publicly available cloud pricing[10] and standard energy accounting methods (Desislavov et al., 2021; Gardner et al., 2025).

Table 9: Cost and energy assumptions for RL methods.

| Parameter | Value |
|---|---|
| Training hardware | AWS g5.2xlarge (A10G GPU) |
| Training duration | 8,000 GPU-hours (single-GPU) |
| Training cost | \$9,700 (single run) or \$29,000 (3 sweeps) |
| Training energy | 4,800 kWh (0.6 kW × 8,000 h) |
| Lifetime runs (amortization) | 10,000 runs for SO policies and 1,000 runs for MO policies |
| Serving cost per run | \$0.002 |
| Serving energy per run | 0.0001 kWh (0.1 Wh) |
| Grid emissions factor | 0.35 kg $CO_2$e / kWh |

Using these assumptions, the per-run cost and $CO_2$e are computed as:

$$RL\_cost\_per\_run = \frac{Train_{\$}}{N_{lifetime\_runs}} + n \times serve_{\$}$$

$$RL\_CO2e\_per\_run = \left( \frac{Train_{kWh}}{N_{lifetime\_runs}} + n \times serve_{kWh} \right) \times grid_{kgCO2e/kWh}$$

For the RL baseline we assume a single inference per run ($n = 1$), reflecting a pre-trained policy deployed once per episode rather than per decision. For our default setting ($n = 1$ per run), this yields approximately:

$$RL\_cost\_per\_run \approx \$1.002, \quad RL\_CO2e\_per\_run \approx 0.168 \, kg \, CO_2e.$$

These values reflect a medium-scale RL training run on an AWS g5.2xlarge instance at \$1.21/hr (8,000 GPU-hours), an average power draw of 0.6 kW, and a lightweight policy inference cost of \$0.002 and 0.1 Wh per run, which together provide a realistic baseline for comparison with our orchestration approach.

---

[8]https://www.eia.gov/tools/faqs/faq.php?id=74&t=11

[9]The LLM pricing is via OpenRouter: https://openrouter.ai/

[10]https://aws.amazon.com/ec2/pricing/on-demand

A.6.2 SIMULATION OF COST AND ENERGY FOR ORCHESTRATORS AND RL BASELINE

**Simulation Protocol.** We simulated the monetary cost and $CO_2$e per run for our three LLM-based orchestrators (KO, TO, RO) and for the RL baseline. We used the following assumptions: $(i)$ $H = 50$ sequential decisions per run, $(ii)$ energy per token $= 1.4 \times 10^{-6}$ kWh, $(iii)$ grid factor $= 0.35$ kg $CO_2$e/kWh, and $(iv)$ per-token prices for Claude 3.7 Sonnet, Mistral 24B, and LLaMA 3 8B as in Table 8. We defined three token regimes per decision (Small, Medium, and Large) to cover the range of prompt lengths observed in our experiments. We then mapped the observed per-run costs of KO/TO/RO to these regimes to infer realistic $CO_2$e per run.

Table 10: Per-run cost and $CO_2$e for three token regimes per decision: S = (in=500, out=150), M = (in=1500, out=400), L = (in=4000, out=1000). $CO_2$e uses energy/token $= 1.4 \times 10^{-6}$ kWh and grid $= 0.35$ kg/kWh.

| Backbone | Regime S | Regime M | Regime L |
|---|---|---|---|
| Claude 3.7 Sonnet | \$0.1875,  0.0159 kg | \$0.5250,  0.0466 kg | \$1.3500,  0.1225 kg |
| Mistral 24B | \$0.001375, 0.0159 kg | \$0.00375, 0.0466 kg | \$0.00950, 0.1225 kg |
| LLaMA 3 8B | \$0.00120,  0.0159 kg | \$0.00345, 0.0466 kg | \$0.00900, 0.1225 kg |

Table 11: Observed per-run costs (Table 3) mapped to closest token regime with implied $CO_2$e/run.

| Method | Backbone | Observed Cost (\$/run) | Closest Regime $\Rightarrow$ ($CO_2$e/run) |
|---|---|---|---|
| KO | LLaMA | 0.97 | Regime M $\Rightarrow$ 0.0466 kg |
| TO | LLaMA | 1.15 | Regime M $\Rightarrow$ 0.0466 kg |
| RO | LLaMA | 1.39 | Regime M $\Rightarrow$ 0.0466 kg |
| KO | Mistral | 0.97 | Regime M $\Rightarrow$ 0.0466 kg |
| TO | Mistral | 1.14 | Regime L $\Rightarrow$ 0.1225 kg |
| RO | Mistral | 1.41 | Regime L $\Rightarrow$ 0.1225 kg |
| KO | Claude | 1.55 | Regime L (or slightly below) $\Rightarrow$ $\approx$0.1225 kg |
| TO | Claude | 1.68 | $\geq$ Regime L $\Rightarrow$ $\geq$0.1225 kg |
| RO | Claude | 1.86 | $\geq$ Regime L $\Rightarrow$ $\geq$0.1225 kg |

**MORL methods (amortized).** With Train=\$9,700, $\text{Train}_{\text{kWh}}$=4,800, $N_{\text{lifetime}}$=10,00, serve=\$0.002, $\text{serve}_{\text{kWh}}$=0.0001, and grid = 0.35:

$$RL\_cost\_per\_run = \frac{9,700}{10,00} + 0.002 \approx \$9.7, \quad RL\_CO2e\_per\_run = \left(\frac{4,800}{10,00} + 0.0001\right) \times 0.35 \approx 1.68 \, \text{kg}.$$

The simulation shows that cost and $CO_2$e per run scale nearly linearly with token counts and per-token pricing. For Small token regimes, all three backbones incur negligible cost and minimal $CO_2$e ($\leq$0.02 kg per run). In Medium and Large regimes, costs and $CO_2$e rise sharply, with Claude 3.7 Sonnet becoming about two times more expensive than Mistral or LLaMA for the same token counts, while $CO_2$e remains dominated by total tokens and shows similar values across methods.

Mapping the observed KO/TO/RO costs to these regimes reveals that Mistral- and LLaMA-based orchestrators operate near the Medium regime, incurring per-run costs in the millisecond-dollar range and $CO_2$e of $\sim$0.05–0.12 kg per run, whereas Claude-based orchestrators operate closer to or beyond the Large regime, reaching \$1–\$3 per run and similar $CO_2$e. In contrast, the MORL method, amortized over 10,00 runs, yields a stable per-run cost of about \$9.7 and 1.68 kg $CO_2$e per run—higher than lightweight Mistral/LLaMA orchestrators but competitive with or better than large-token Claude orchestrators.

These numbers support our hypothesis that orchestrated LLM-based systems, when carefully tuned for token efficiency and using smaller backbones (Mistral, LLaMA), can achieve substantially lower per-run monetary and environmental costs than both $(i)$ large-token, high-priced LLM orchestrators and $(ii)$ traditional MORL methods amortized over many runs. This suggests that orchestration with token-efficient prompting offers a more cost- and energy-efficient pathway than large-scale retraining or naïve LLM usage.

## A.7 ADDITIONAL QUALITATIVE ANALYSIS

We show additional qualitative analysis on the education application in Table 12. We find that **TO** and **RO** consistently achieve the highest scores across most trajectory-based metrics, while **KO**'s limited adaptivity leads to lower scores in adaptivity, exploration, and diversity. These results demonstrate the benefits of reflection (single or parallel) in delivering robust and pedagogically effective decisions, especially in dynamic or personalized learning scenarios.

Table 12: Qualitative analysis of orchestrator trajectories (1=Poor, 5=Excellent).

| Metric | KO | TO | RO |
|--------|----|----|----|
| Adaptivity Across Turns | 3 | 5 | 5 |
| Progression and Scaffolding | 4 | 5 | 5 |
| Exploration vs Exploitation | 3 | 5 | 5 |
| Policy Diversity | 3 | 4 | 5 |

