# OpenReview forum: "Orchestrating Pre-Trained Agents for Multi-Objective Decision Making"
_ICLR.cc/2026/Conference — Submitted to ICLR 2026_

### Official Review · Reviewer_wH1w · 2025-10-17

**Soundness:** 2
**Presentation:** 2
**Contribution:** 1
**Rating:** 2
**Confidence:** 4

**Summary:**

The paper proposes "orquestration" strategies in which an LLM is used to select RL policies with the intention of improving multi-objective RL. The paper dives into details of how to build the context of the LLM for this specific task and shows experimental results in a mock "learning" environment showing how the LLM fares against simple baselines using scalarization.

**Strengths:**

The paper is very detailed in how the context is built for the experiments.

Paper is well-written and easy to understand

Paper took care of evaluate costs of running the different models in the performance evaluation

**Weaknesses:**

- The major weakness that resulted in the evaluation I am giving to this paper is that the setting of the problem the paper is trying to solve makes absolutely no sense to me and feels like this paper is an attempt of force-feeding LLMs into multi-objective RL in a way that it is not the best solution for that. I cannot think of a single practical scenario where we would have available numerous single-objective RL policies pre-trained and would want to use those policies to solve a multi-objective problem. If we are dedicating the time to train the policies one would straight away use a pareto-optimizing RL algorithm, instead of optimization it many times in a single objective manner. A scenario that *maybe* would make sense to me is that we want a solution with configurable scalarization weights, so you cannot optimize for the weights beforehand because they are unknown. In this scenario - maybe - we could assume that a large collection of policies with varying combinations of weights would be available and the LLM would be in charge of selecting which one fits best with the preference of the user given in plain english. But this scenario is very different from the one presented in the paper.

- The environment used for validation is an unusual one and it doesn't seem like one could extract meaningful insights from it without human experiments. If the intention was to just have a toy problem, it would have been better to use a collection of different environments from for example MO-gymnasium https://github.com/Farama-Foundation/MO-Gymnasium

- The results presented look weird, how did the baselines resulted in order of magnitude higher costs when they can very easily be executed in the user's own CPU/GPU? furthermore, the LLM-orchestrated runs also have to execute the RL envrionments in the same way to solve the problem, so there shouldn't be much saving from using the LLM even without considering the cost per token.

- I also don;t understand why the metrics are presented the way they are in the evaluation - it would be much simpler and more meaningful the simply showing hypervolume and sparsity.

**Questions:**

No specific questions

---

> ### Author Response · Authors · 2025-11-26
>
> **W1.** We appreciate the reviewer’s perspective and the opportunity to clarify the intended scope of our setting. Our goal is *not* to replace classical MORL pipelines, nor to suggest that single objective (SO) policy orchestration is universally preferable. Rather, our work targets a **specific but important class of real-world scenarios** where the objective set is uncertain or changes across contexts. For instance, in education applications, strong students may benefit from questions that maximize expected performance *and* learning potential, and weaker students may require content that maximizes expected performance *and* reduces knowledge gaps to maintain their engagement. Additionally, new objectives may emerge as environments evolve. In such scenarios, retraining MO policies becomes prohibitively expensive.
>
> Many deployed RL systems (e.g., education, healthcare triage, recommender pipelines, industrial robot control) maintain specialized pre-trained SO policies that were previously trained and validated. Our work seeks to **make the best use of those existing SO policies** to achieve multi-objective outcomes.
>
> Our contribution is to explore whether **language models can leverage their world knowledge and context representations to orchestrate existing SO policies**, enabling low-cost multi-objective decision-making when retraining is impractical. We acknowledge that this is a more specialized scenario than classical MORL, and we have revised **Section 1 Introduction** to make this scope explicit.
>
> ---
>
> **W2.** Our education environment was chosen because it is inherently multi-objective (aptitude, performance, gap) and mirrors real tutoring-system trade-offs. However, we agree that validating on a standard MORL benchmark increases generality. Hence, we have **added new experiments using a MiniGrid environment from MO-Gymnasium** in **Section 4 Experiments**. We thank the reviewer for this opportunity to show generalizability.
>
> ---
>
> **W3.** We appreciate the opportunity to clarify our cost assumptions. First, it is important to note that for both multi-objective RL (MORL) and single-objective RL (SORL) methods, the **training cost dominates the total cost**, due to expensive environment interactions, sample collection, and gradient updates. In contrast, the inference cost is negligible in typical RL settings and is therefore omitted in our cost analysis for MORL/SORL.
>
> - **MORL methods require training from scratch for each objective configuration.** Training MO-PPO or MO-A2C demands tens of thousands of environment steps and repeated gradient updates.
> For our orchestration-based method, the cost structure is different:
> - If SO policies exist, orchestration only incurs token-level LLM inference cost.
> - If SO policies must be trained, their cost can be amortized across future reuse.
>
> We added a new table to **Section 4 Experiments**. It can be found in the response to Reviewer 1 W6 / Q5.
>
> ---
>
> **W4.** Thank you for this suggestion. In the revised version, we have added hypervolume and sparsity evaluations for all methods to complement the original per-objective and task-specific metrics.

---

### Official Review · Reviewer_tvPz · 2025-10-20

**Soundness:** 2
**Presentation:** 3
**Contribution:** 2
**Rating:** 4
**Confidence:** 4

**Summary:**

This paper proposes novel approaches for reusing pre-trained RL policies to solve multi-objective decision-making tasks via an orchestration mechanism in LLMs. The proposed technique is based on the context engineering formalization by Meietal. (2025), and the main idea is that the orchestration is able to select the most appropriate base RL policy based on three different types of contextual information: Knowledge-based Orchestrator (KO), Tool-based Orchestrator (TO), and Reflection-based Orchestrator (RO). These three approaches are evaluated in an application in Education, where the goal is to recommend questions to students to attain skill mastery. In this scenario, there are three conflicting objectives: aptitude, gap, and performance. The method is compared with single RL policies optimized via scalarization of objectives or via reward machines.

**Strengths:**

- Multi-objective decision-making is a very relevant problem, and building intelligent agents able to control trade-offs between conflicting rewards is challenging.

- The experiments are very extensive and were conducted using an insteresting and complex application. It discusses many different metrics/advantages and properties of each approach.

**Weaknesses:**

- The method is an application of existing context engineering formalizations (Mei et al. (2025)), and it is not very clear if the only theoretical contribution is the application of such techniques to a multi-objective task.

- The objective of the paper is posed in a misleading way. The research question of the paper is stated as:  “Can we combine existing single objective policies to better cover the Pareto frontier?”
However, the proposed approach is not trying to cover the Pareto frontier. The orchestrators are learning a single policy that obtains a reasonable trade-off between all Pareto-optimal solutions. The method is not trying to cover all Pareto optimal solutions, but only to achieve a single one. How would it be possible to generate multiple solutions in the Pareto front based on dynamic user preferences via the proposed approach?

- A vast literature on multi-objective RL (MORL) algorithms exists, but was ignored in this paper. See [1-4] below, for instance. These methods should have been discussed, since they intend to approximate the Pareto frontier by learning multiple policies separately or by learning a single policy conditioned on preferences that generalizes over multiple weights. The authors, however, only considered a single-objective baseline that uniformly weights all objectives. It would be very relevant to observe how these approaches would compare in the experiments. There are also MORL approaches in particular applied to LLMs that need to be discussed [5-7].

- There must be a misunderstanding, but I do not see how the cost calculation is computed in a fair way. The orchestrators required a set of pre-trained RL policies, and at inferece time they select one of the available policies, as state in the beginning of Section 3.2. Hence, the cost of the orchestrator approach (e.g., TO) involves the cost of inference of each of the pre-trained candidate RL policies. The multi-objective scalarization policy (MO-S), however, only involves training and inference of a single RL policy. It is completly unclear to me how an orchestrator can be less costly than a pure RL approach if it also involves training RL-policies and selecting one of then at inference time.

- The last section of the paper is Section 4 - Experiments. The paper is missing a Conclusion section to summarize the main findings, discuss the limitations and future works.

[1] Alegre, Lucas N., Ana L. C. Bazzan, Diederik M. Roijers, Ann Nowé, and Bruno C. da Silva. ‘Sample-Efficient Multi-Objective Learning via Generalized Policy Improvement Prioritization’. Proceedings of the 2023 International Conference on Autonomous Agents and Multiagent Systems.

[2] Hayes, Conor F., Conor F. Hayes, Rădulescu, Roxana, et al. ‘A Practical Guide to Multi-Objective Reinforcement Learning and Planning’. Autonomous Agents and Multi-Agent Systems 36, no. 1 (2021). https://doi.org/10.1007/s10458-022-09552-y.

[3] Röpke, Willem, Mathieu Reymond, Patrick Mannion, Diederik M Roijers, Ann Nowé, and Roxana Radulescu. ‘Divide and Conquer: Provably Unveiling the Pareto Front with Multi-Objective Reinforcement Learning’. Proceedings of the 24th International Conference on Autonomous Agents and Multiagent Systems.

[4] Lu, Haoye, Daniel Herman, and Yaoliang Yu. ‘Multi-Objective Reinforcement Learning: Convexity, Stationarity and Pareto Optimality’. The Eleventh International Conference on Learning Representations. 2022. https://openreview.net/forum?id=TjEzIsyEsQ6.

[5] Wang, Kaiwen, Rahul Kidambi, Ryan Sullivan, et al. ‘Conditional Language Policy: A General Framework for Steerable Multi-Objective Finetuning’. arXiv:2407.15762. Preprint, arXiv, 23 October 2024. https://doi.org/10.48550/arXiv.2407.15762.

[6] He, Qiang, and Setareh Maghsudi. ‘Pareto Multi-Objective Alignment for Language Models’. arXiv:2508.07768. Preprint, arXiv, 11 August 2025. https://doi.org/10.48550/arXiv.2508.07768.

[7] Jafari, Yasaman, Dheeraj Mekala, Rose Yu, and Taylor Berg-Kirkpatrick. ‘MORL-Prompt: An Empirical Analysis of Multi-Objective Reinforcement Learning for Discrete Prompt Optimization’. Findings of the Association for Computational Linguistics: EMNLP 2024, Association for Computational Linguistics, 2024, 9878–89. https://doi.org/10.18653/v1/2024.findings-emnlp.577.

**Questions:**

- In line 130, the authors define the return variable without the discount factor $\gamma$. However, in Table 5 the value of $\gamma$ used by the algorithms is set to 0.99. Did the authors evaluate if there is any discrepancy between discounted and undiscounted returns?

- Why is the MDP constrained to be deterministic?

- “We model the environment as a multi-reward MDP” ->
In the multi-objective RL, this is called a multi-objective MDP (MOMDP). I would suggest using this nomenclature to be consistent with the rest of the literature.

- For completeness and improved clarity, I suggest discussing and presenting the Context engineering formalization by Mei et al. (2025) in more detail at the beginning of Section 3.1.

- “For MOO, we modified the training of A2C and PPO by introducing a replay buffer that stores trajectories from only non-dominated episodes.”
Could you please elaborate on why this was necessary? Known applications of PPO for MORL do not implement such filtering in the replay buffer.

- In Appendix A.2, the authors express the SARSA update equation for tabular RL. Since the authors used neural networks as function approximation, I suppose that a DQN-style loss was used instead? Please clarify.

- “We align TO with the Reflexion Paradigm Shinnetal. (2023) which models a single linear test time scaling loop with three roles: an actor (performing the decision), an evaluator (assessing the quality of the decision given the environment), and a self-reflection step (feeding back into the next decision).”
How were these implemented? How does the evaluator assess the quality of the decision? Since this is a multi-objective decision-making policy, the quality is subjective to the user preferences. How are the preferences inferred?

Minor:
- The authors are using \citet when the correct would be \citep.

---

> ### Author Response · Authors · 2025-11-26
>
> **W1 / Q4.** Thank you for raising this concern. While we build on context engineering (CE)~\citep{mei2025}, we emphasize that our contribution is not a mere application of CE, but a conceptual extension tailored to the challenges of multi-objective sequential decision-making.
>
> Specifically, we formalize the problem of inference-time policy orchestration under MOMDP using CE:
> $$
> \max_{C} P_\theta(\pi^* \mid C)
> $$
> That is, we aim to maximize the probability that an autoregressive LLM, parameterized by $\theta$, selects the optimal policy $\pi^*$ given a context $C$. This formulation enables zero-shot adaptation to unknown objective configurations, which is not addressed in prior CE work. We added this to **Section 3.1**.
>
> ---
>
> **W2.** We appreciate this important clarification. We agree our method does not aim to **cover** the full Pareto frontier, but rather to obtain reasonable trade-offs among conflicting objectives and find *some* Pareto-optimal solutions.
>
> We have revised the phrasing of the central question and clarified in **Section 2** that our goal is to dynamically select among existing single-objective policies to achieve a competitive trade-off across conflicting objectives, without retraining.
>
> Extending our method to preference-aware orchestration or multi-solution generation is an exciting direction that we will highlight in our (newly added) conclusion as future work.
>
> ---
>
> **W3.** Thank you for pointing this out. We expand our related work discussion to review both classical MORL (scalarization, Pareto, gradient-based) [1–4] and MORL methods that specifically apply to optimizing LLMs [5–7] in **Appendix A.1**.
>
> Our difference from classical MORL and RL reuse methods is that we aim to operate in settings where **objectives are uncertain or context-dependent**, and thus **training from scratch or retraining is prohibitively expensive**. Compared to methods that apply RL to LLMs, our focus is instead on whether **LLMs can orchestrate or reuse SO policies** to find results that are comparable to MO policies (in **MOMDP** settings, not in language model training).
>
> To address the comparison gap, in **Section 4 Experiments**, we also added experiments with Pareto-based and gradient-based variants. Results show our approach matches or outperforms MO baselines on **hypervolume** and **sparsity**.
>
> ---
>
> **W4.** Thanks for the opportunity to clarify cost assumptions. First, in both MORL and SORL, **training dominates cost** due to env. interactions and optimization. Inference cost is negligible.
>
> In our framework:
> - If SO policies exist, orchestration only incurs token-level LLM inference cost.
> - If SO policies must be trained, their cost can be amortized across future reuse.
>
> We added a new table to **Sec. 4**. It can be found in the response to Reviewer 1 W6 / Q5.
>
> ---
>
> **W5.** Thank you, we will add one. This will also be an opportunity for us to discuss extending our method to preference-aware orchestration or generating multiple Pareto-optimal solutions (e.g., through conditional prompting).
>
> ---
>
> **Q1.** Thank you, you are correct. We have revised the formula, and we confirm that all implementations use $\gamma=0.99$, consistent with Table 5.
>
> ---
>
> **Q2.** We focus on deterministic MDPs to limit the complexity and scope of the study while still evaluating the core research questions around inference-time orchestration, cost efficiency, and multi-objective trade-offs. Both MathE and MiniGrid are deterministic. Extending to stochastic MDPs is now discussed in the conclusion.
>
> ---
>
> **Q3.** We have updated the terminology.
>
> ---
>
> **Q5.** We apologize for the confusion. In the current version of the paper, **we do not use replay buffers** for PPO and A2C. We will update the Appendix to reflect this.
>
> ---
>
> **Q6.** We confirm our SARSA implementation uses a neural Q-function and an on-policy TD target (not DQN loss). This is now clarified in the Appendix.
>
> ---
>
> **Q7.** We thank the reviewer for this insightful question. Our TO orchestration follows the Reflexion loop:
>
> - **Actor**: The LLM orchestrator receives a structured prompt (including `c_query`, `c_state`, and `c_mem`) and selects a single-objective policy $\pi_i$ from the policy zoo to execute.
> - **Evaluator**: The selected policy $\pi_i$ is executed in the environment for $k$ steps ($k=1$ for TO). The environment returns an outcome (e.g., mastery level, corrected questions, failed questions), which is summarized into a structured JSON.
> - **Reflection**: This structured JSON is inserted into `c_mem`, which is reused in the next round of orchestration. The LLM can then decide to (i) call another policy to observe its behavior, or (ii) terminate the orchestration loop by selecting a final policy.
>
> Regarding **preferences**, these are encoded in `c_query`/`c_instr` and assumed given per task. We will clarify this in the Appendix.

---

### Official Review · Reviewer_sRAN · 2025-10-30

**Soundness:** 2
**Presentation:** 2
**Contribution:** 2
**Rating:** 2
**Confidence:** 2

**Summary:**

This work proposes a model that reuses pre-trained RL policies rather than retraining them for each new tasks in multi-objective decision making, and applies the model to an education system.

**Strengths:**

Motivation is clear - making use of LLM to assist in reusing pre-trained RL policies sounds reasonable.

**Weaknesses:**

It is not clear to me why you want to considier multi-objective decision making - using LLM to assist pre-trained RL policies does not apply to the mono-objective case? It seems to me that the method does not make full use of the multi-objective case, except considering nondominated policies.

I am not an expert in multi-objective RL, but you stated that policy gradients are computed only from steps belonging to these non-dominated trajectories. I am wondering how this could be done. Since you may have many incomparable policies, how you compute gradients without aggregating them? Especially, in the case there are many objectives, almost all policies could be incomparable.

There are no much descriptions and explanations about the figures - I feel difficulty to understand them. Figures should be self-contained - without checking the text, readers should be able to understand them (e.g. via legends and captions).

Typo: ``...on an non-overlapping subset...'' in Figure 1.

**Questions:**

Could you please respond to the first two comments in the weaknesses field?

---

> ### Author Response · Authors · 2025-11-26
>
> **W1 / W2.** We answer both weaknesses together as they are intertwined.
>
> There are several applications where optimizing more than one objective is needed: a student test recommendation system seeks to optimize expected performance *and* learning potential; a robot traverses a grid and seeks to reach “target “ quickly *and* and to collect as much gold as possible *and* to avoid dangerous cells. We now address both applications (and their empirical validation) in the paper.
>
> In this work, we do not train RL policies. We seek to reuse them in **scenarios where the objective set is uncertain or changes with context**, which makes retraining multi-objective (MO) RL policies prohibitively expensive. To avoid expensive retraining, we explore whether **language models can act as inference-only orchestrators** to reuse single-objective (SO) policies for MO decision-making. We revisited our introduction to review the relevant literature and our positioning (**Section 1 Introduction**).
>
> Our empirical validation shows that reusing pre-trained SO policies is a good idea and produces results that are comparable and even superior to MO policies without the need for retraining.
>
> ---
>
> **W3.** For the figures, we have revisited all figure captions to make them self-contained.
>
> ---
>
> **W4.** Thanks for pointing out the typo and we have fixed it.

---

### Official Review · Reviewer_LGoB · 2025-11-01

**Soundness:** 2
**Presentation:** 2
**Contribution:** 2
**Rating:** 2
**Confidence:** 4

**Summary:**

The paper proposes a novel approach to multi-objective sequential decision-making by orchestrating pre-trained single-objective RL policies using large language models (LLMs). The goal is to avoid retraining costs while leveraging LLMs’ natural language reasoning capabilities to choose from a set of expert policies under different objective priorities. The approach is evaluated primarily in a newly introduced educational recommendation environment, with experiments spanning scalarized objectives, LLM guidance, and human-judgment comparisons.

While the idea of post-training orchestration is timely and potentially impactful, the paper lacks sufficient theoretical and empirical grounding, particularly with respect to existing multi-objective reinforcement learning (MORL) literature and metrics. Several conceptual components (e.g., prompts, agents, contexts) also lack integration into a cohesive framework

**Strengths:**

- The paper explores an original direction, zero-shot multi-objective decision-making by combining existing policies via LLM guidance, addressing the cost and scalability challenges common in RL.
- The emphasis on energy and computational savings aligns with pressing concerns around sustainable AI.
- The authors provide a detailed experimental setup and introduce a domain-specific benchmark environment that may be of value to the community if released openly.
- The modular design of the orchestrators (knowledge-based, tool-based, reflection-based) reflects creative thinking toward policy composition.

**Weaknesses:**

-The paper overlooks foundational work in Multi-Objective Reinforcement Learning (MORL), where many of the addressed themes, trade-off learning, policy reuse, Pareto approximation, have already been studied in detail. Similarly, although the approach resembles zero-shot policy composition, the authors do not employ this terminology nor connect to existing zero-shot RL or policy reuse work, which raises concerns about positioning and familiarity with relevant paradigms.

- The central idea, composing pre-trained single-objective policies via LLMs for multi-objective tasks, is not sufficiently justified over established MORL techniques, which are capable of learning diverse trade-offs directly. Relying on fixed single-objective policies may also inherently limit Pareto coverage.

- The paper introduces important components (prompts, agents, context embeddings) in isolation without a global design blueprint. This makes it difficult to understand how the system functions holistically or to generalize it beyond the educational domain.

- The experiments assess performance only using scalarized metrics, without considering standard multi-objective measures like hypervolume or Pareto cardinality. As a result, empirical claims about effectively handling trade-offs remain insufficiently demonstrated.

- The approach is evaluated only on a single, synthetic domain and relies heavily on textual descriptions of policies. It is unclear how well this setup would generalize to higher-dimensional state spaces or control tasks where policy semantics are not easily verbalized.

- It is not fully transparent whether the cost and CO₂ analyses include the initial single-objective RL policy training stages. If not, this skews the claimed “carbon efficiency” in favor of the proposed method.

**Questions:**

- How does your approach compare to MORL methods that directly learn diverse policies or Pareto fronts, such as evolutionary MORL, preference-conditioned policies, or hypernet-based approaches? Could your framework also be interpreted as a form of zero-shot RL or post-training policy reuse?

- Why were canonical multi-objective metrics such as hypervolume or ε-indicator not incorporated into the evaluation? Would including these metrics change your conclusions about policy coverage or performance?

- Have you considered testing the orchestration approach on standard multi-objective RL benchmarks in robotics or resource allocation, where action spaces and objectives are more complex?

- How do performance and cost scale as the number of pre-trained policies increases, or when policies are not easily described in textual form?

- Does your cost estimate include all life-cycle costs of the RL policies being reused (i.e., training time and compute)? If not, can you clarify the assumptions behind your analysis?

- Will the educational benchmark environment and orchestration code be made available as open-source contributions to the community?

---

> ### Author Response · Authors · 2025-11-26
>
> **W1 / Q1.** We thank the reviewer for highlighting this important positioning issue. For MOO tasks, we agree that foundational MORL literature is of very high relevance. We have added a new **Appendix A.1: Related Work** section that surveys MORL approaches including scalarization-based, Pareto-based, and gradient-based methods. All these methods require training once the set of objectives are known. Our work differs in its assumptions: we seek to leverage pre-trained single-objectives policies.
> Our work focuses on **scenarios where the objective set is uncertain or changes with context**, which makes retraining prohibitively expensive. Hence, our work is more closely related to **RL policy reuse**, which we now also review in A.1. That said, most existing reuse methods (e.g., meta-RL) still require costly retraining procedures.
> To avoid expensive retraining, we explore whether **language models can act as inference-only orchestrators** to reuse single-objective (SO) policies for multi-objective (MO) decision-making. We also revisited our introduction to review the relevant literature and our positioning (**Section 1 Introduction**).
> To the best of our knowledge, ours is the first work that performs inference-only policy reuse via LLMs.
>
> ---
>
> **W2.** Our core motivation is that when **objectives are uncertain or context-dependent**, retraining a new MO policy is costly. In contrast, SO policies are cheaper and often already available. Our work explores whether **language models can leverage their world knowledge and context representations to orchestrate SO policies** and finds solutions that are comparable to Pareto-based approaches.
> We strengthened this in **Section 1**, and added Pareto- and gradient-based MO baselines in **Section 4**. Results show that orchestrating fixed SO policies achieves comparable or even superior hypervolume and lower sparsity than retrained MO policies.
>
> ---
>
> **W3.** We agree that a cohesive global design is critical. We revised **Section 3.1**, where we formalize orchestration as:
> $$
> \max_{C} P_\theta(\pi^* \mid C)
> $$
> That is, we aim to maximize the probability that an autoregressive LLM, parameterized by $\theta$, selects the optimal policy $\pi^*$ given a context $C$. As in the general formulation of context engineering:
> $$
> C = \mathcal{A}(c_1, c_2, \dots, c_n)
> $$
> where each $c_i$ is a context component sourced, filtered, and formatted by a set of functions, then assembled by $\mathcal{A}$.
> In our orchestration setting, the components $\{c_i\}$ directly map to the needs of MO sequential decision-making, such as `c_query`, `c_state`, `c_instr`, etc, which we have explained in Section 3.1.
>
> ---
>
> **W4 / Q2.** We have computed **hypervolume** and **sparsity** and included them in our empirical validation.
>
> ---
>
> **W5 / Q3 / Q4.** We agree that broader validation is essential. We added **MiniGrid from MO-Gymnasium**, whose 147D state space extends beyond the 7D educational setting. This shows that our method generalizes to higher-dimensional environments.
> Regarding the generalizability when policy semantics are not easily verbalized, it is important to clarify that only KO relies on natural-language descriptions of policy behavior. TO and RO treat policies as **black-box callable modules**, requiring no natural-language semantics. This allows our framework to be applied in domains such as robotics, navigation, or resource allocation, where policy semantics are opaque or hard to verbalize.
>
> ---
>
> **W6 / Q5.** Thank you for the question. To clarify:
> - Both MORL and SORL approaches have a training cost and a negligible inference cost.
> - If SO policies are already available, the orchestrator incurs only inference cost.
> - If SO policies must be trained, their cost can be amortized over future reuse, as they are reusable under our framework.
> We have added a table in **Section 4 Experiments** comparing **training cost**, **inference cost**, and **total cost** between MORL and our orchestration framework.
> **Table: Computational cost (\$/run) comparison across methods.**
> | Method | Backbone | Training Cost | Inference Cost | Total Cost |
> |--------|----------|----------------|----------------|-------------|
> | SORL   | /        | 0.97           | 0.002          | 0.972       |
> | MORL   | /        | 9.7            | 0.002          | 9.702       |
> | KO     | Mistral  | 0.972          | 0.0037         | 0.976       |
> | TO     | Mistral  | 0.972          | 0.012          | 0.984       |
> | RO     | Mistral  | 0.972          | 0.0091         | 0.981       |
> | KO     | Claude   | 0.972          | 1.14           | 2.072       |
> | TO     | Claude   | 0.972          | 2.96           | 3.932       |
> | RO     | Claude   | 0.972          | 2.45           | 3.422       |
>
> ---
>
> **Q6.** Yes, the environments and orchestration framework (KO/TO/RO prompts, LLM backbones, evaluation scripts) will be open-sourced upon acceptance. Code is included as supplementary material on the submission portal.

---

### Author Response · Authors · 2025-11-29
**Summary of Changes**

**Summary of Changes:**

- **Introduction (50%)**: We clarified that the core problem is multi-objective sequential decision making (MO-SDM) addressed via policy reuse. We strengthened the motivation by emphasizing scenarios where objectives are unknown or context-dependent. We also added a discussion on traditional MORL methods and highlighted key differences from our approach.

- **Section 2 (5%)**: We aligned our terminology with the literature (multi-reward MDP -> MO-MDP) and refined the phrasing of the key challenge.

- **Section 3 (20%)**: We revisited the formalization of the zero-shot policy orchestration problem under the framework of LLMs. We defined context components more precisely and anchored our design in prior work (Mei et al., 2025). The prompt templates were revised to generalize across domains, and we clarified the relationship between Tool-based Orchestration (TO) and the Reflexion work (Shinn et al., 2023).

- **Section 4 (50%)**: Following reviewers’ suggestions on generalizing the applicability of our work, we introduced a new application (MiniGrid) and updated the evaluation to include Pareto front quality metrics (hypervolume and sparsity). We also highlighted that reference MO policies provide performance upper bounds, and added MO-MGDA as a new reference MO policy alongside MO-S and MO-RM. MiniGrid results and analysis are now added to Sections 4.2.1 and 4.2.2. Our empirical findings on MiniGrid corroborate our findings on the Education use case.

- **Section 5**: This section was newly added.

---

### Meta-Review · Area_Chair_pNb1 · 2026-01-05

**Summary:**

Reviewers generally like the problem and think that the proposed approach interesting, but also raised concerns on insufficient theoretical or empirical justifications, and insufficient discussion and comparisons with previous work.

**Reviewer Concerns:**

see above.

**Reviewer Scores:**

N/A

---

### Decision · Program_Chairs · 2026-01-26

Reject